# Weak Diffusion Priors Can Still Achieve Strong Inverse-Problem Performance

**Jing Jia** [* 1]  **Wei Yuan** [* 2]  **Sifan Liu** [3]  **Liyue Shen** [4]  **Guanyang Wang** [2]

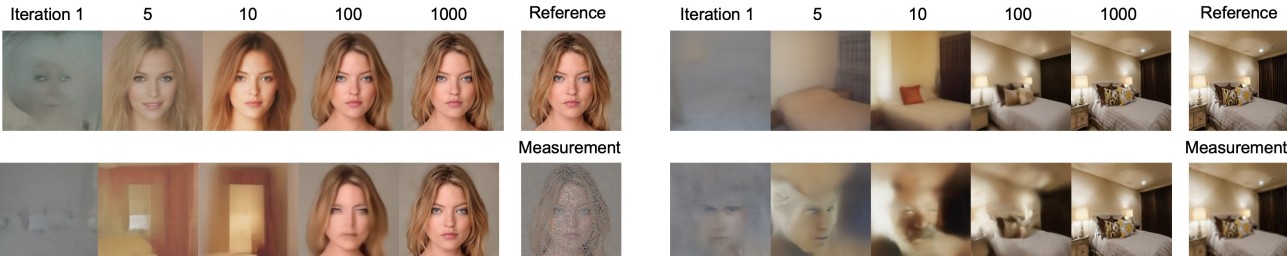

*Figure 1.* Image reconstruction using only a 3-step DDIM generator as the prior. Left block: reconstruction of a face image; right block: reconstruction of a bedroom image. In each block, the top row uses a matched prior, while the bottom row uses a mismatched prior. Diffusion models trained on bedroom images can still reconstruct face images, and diffusion models trained on face images can still reconstruct bedroom images. From left to right, we show intermediate reconstructions over optimization iterations. The Reference column shows the clean image, and the Measurement column shows the noisy observation.

## Abstract

Can a diffusion model trained on bedrooms recover human faces? Diffusion models are widely used as priors for inverse problems, but standard approaches usually assume a high-fidelity model trained on data that closely match the unknown signal. In practice, one often must use a mismatched or low-fidelity diffusion prior. Surprisingly, these weak priors often perform nearly as well as full-strength, in-domain baselines. We study when and why inverse solvers are robust to weak diffusion priors. Through extensive experiments, we find that weak priors succeed when measurements are highly informative (e.g., many observed pixels), and we identify regimes where they fail. To explain this behavior, we combine Bayesian-consistency theory with local-correlation analysis: the theory gives conditions under which high-dimensional measurements make the posterior concentrate near the true signal, while the correlation analysis shows that weak and stronger natural-image priors can share similar local spatial structure. These results provide a principled justification on when weak diffusion priors can be used reliably. Code is available at this repository.

## 1. Introduction

Inverse problems aim to recover an unknown signal from noisy measurements. Recently, diffusion models have emerged as powerful *data-driven priors* for inverse problems, achieving state-of-the-art performance across a wide range of tasks in computer vision and related fields. The standard paradigm employs the same "full-strength" diffusion model as the prior: for example, a 1000-step DDPM (Ho et al., 2020; Song et al., 2021b) trained on a face dataset to recover human faces. Building on this, existing solvers integrate the diffusion model directly into reconstruction by injecting measurement information throughout a reverse diffusion chain to enforce consistency with the observation (Song et al., 2021b; 2022; Chung et al., 2023; Kawar et al., 2022; Meng & Kabashima, 2024; Zhou et al., 2024; Song et al., 2024). This strategy constitutes the mainstream approach for diffusion-based inverse problem solving.

However, the luxury of a perfectly matched, full-strength prior is often unavailable. Practical constraints often force us to use "weaker priors," meaning diffusion models that are heavily truncated at inference or trained on a mismatched

---

[*]Equal contribution  [1]Department of Computer Science, Rutgers University, Piscataway, United States [2]Department of Statistics, Rutgers University, Piscataway, United States [3]Department of Statistical Science, Duke University, Durham, United States [4]Department of EECS, University of Michigan, Ann Arbor, United States. Correspondence to: Jing Jia <jing.jia@rutgers.edu>, Guanyang Wang <guanyang.wang@rutgers.edu>.

*Proceedings of the $43^{rd}$ International Conference on Machine Learning*, Seoul, South Korea. PMLR 306, 2026. Copyright 2026 by the author(s).

dataset. For example, Wang et al. (2024) and Chihaoui et al. (2024) propose optimizing the initial noise to solve inverse problems. To meet the memory constraints, they have to drastically truncate the diffusion process to just a few steps. Consequently, the prior itself is severely limited: it can only generate low-fidelity, blurry images (see `Iteration 1` in Figure 1). In parallel, in data-scarce domains such as medical image restoration, the absence of training data precludes the creation of domain-specific models. Therefore, researchers often compromise by relying on pre-trained models from completely mismatched distributions (Knoll et al., 2019; Jalal et al., 2021; Glaszner & Zach, 2024; Aali et al., 2025; Hu et al., 2025; Barbano et al., 2025).

Surprisingly, these weak priors can still yield highly competitive reconstructions. For example, Wang et al. (2024) reports a 2–6 dB PSNR improvement over state-of-the-art baselines across a range of inverse problems, despite using only a 3-step DDIM prior (Song et al., 2021a). Likewise, Jalal et al. (2021) shows that a diffusion prior trained exclusively on 2D brain MRI can still be used to reconstruct out-of-distribution scans such as knee and abdomen MRI.

These results raise a puzzle: standard intuition suggests that the quality of the reconstruction is bounded by the quality of the prior. Yet, in these regimes, weak priors perform nearly as well as, or even better than, stronger ones. At present, such successes are largely reported case by case in application-driven studies, and a systematic understanding of *when* and *why* weak priors suffice remains lacking.

Motivated by these observations, we ask: *can a low-fidelity bedroom diffusion model recover face images?* More broadly, we study when an inverse problem is robust to the choice of prior, in the sense that a low-quality or mismatched generative prior can still yield high-quality reconstructions, and when it cannot.

Our main contribution is a rigorous characterization of this phenomenon, supported by empirical experiments, theoretical analysis, and local-correlation diagnostics. **Our central message** is that weak priors can perform strongly in data-informative regimes. This happens because of two related effects. First, data can dominate the prior: when the measurement has large effective dimension, reconstruction is driven more by the observation than by prior. Second, weak priors are not as weak as their samples suggest: even few-step or out-of-domain natural-image priors can share local spatial correlations with stronger matched priors. These two effects together provide a *complementary explanation*: informative measurements supply local anchors, while the prior propagates these constraints through shared local spatial structure.

Conceptually, our results show that the prior may matter less than is often assumed when measurements are sufficiently informative: the value of a strong, domain-matched prior depends on measurement informativeness. Practically, this supports using weak priors as a reasonable default in data-informative regimes where strong, domain-matched priors are unavailable, while reserving stronger matched priors for low-information settings where reconstruction is more prior-dependent. To support this, we provide:

1. **Empirical evidence:** We conduct extensive experiments across inverse problems, datasets, and diffusion backbones. We find that even severely truncated or mismatched priors can yield accurate reconstructions in data-informative regimes.

2. **Mechanistic explanation:** We combine Bayesian-consistency theory with local spatial-correlation diagnostics to explain why weak priors can work. The theory links measurement informativeness to reduced prior sensitivity: when the effective observed dimension is large and the signal is identifiable from the measurements, the posterior concentrates near the best measurement-consistent mode and the influence of the prior weights is reduced. The spatial diagnostics show that weak and stronger natural-image priors can share nearly identical local-correlation decay curves, suggesting that even mismatched priors can provide useful local structural information once the measurements supply enough anchors.

3. **Failure modes:** Beyond the successful cases, we study when weak priors fail. These include box inpainting and large-scale super-resolution, where the observation leaves a large missing region or a small effective observed dimension. In these regimes, reconstruction becomes more prior-dependent, and weak or mismatched priors can produce inconsistent structures. We support these findings with both theory and experiments.

4. **Algorithmic refinements:** To realize the prior-robust behavior predicted by our analysis, we refine initial-noise optimization with a new optimizer and an improved early-stopping rule, which stabilize reconstruction and reduce overfitting.

## 2. Weak diffusion generators as priors

### 2.1. Setup and our algorithm

**Formulation:** We study inverse problems of the form $y = \mathcal{A}(x) + \epsilon$, where $y \in \mathbb{R}^m$ denotes the observed data and $x \in \mathbb{R}^n$ is the unknown signal to be reconstructed. The map $\mathcal{A} : \mathbb{R}^n \to \mathbb{R}^m$ is called the *forward operator*, and $\epsilon \sim \mathbb{N}(0, \sigma^2 I_m)$ models additive Gaussian noise.

**Diffusion prior:** The mapping from $x$ to $y$ is typically many-to-one, so recovering $x$ from $y$ is ill-posed. To reg-

ularize the problem, we place a prior distribution $\pi$ on $x$ that captures the structure of plausible signals (e.g., natural images). In modern generative modeling, such a prior is often specified through a pretrained generator $G$ that maps Gaussian noise to a sample in data space. In this paper, we focus on diffusion-based generators, where $G$ is realized by a finite number of reverse diffusion steps (e.g., a $k$-step DDIM sampler).

**Weak prior in two axes:** Most prior work uses a *strong and matched* diffusion prior: a high-quality diffusion sampler, often using the original long reverse chain, trained on a distribution that closely matches the target signals. In this paper, we call a prior *weak* if it lacks one or both of these properties. Thus, weak diffusion priors can arise along two independent axes:

- *Generator quality:* $G$ is obtained by running only a few reverse steps (e.g., a 1–4 step DDIM sampler), which typically yields low-fidelity unconditional samples.

- *Domain match:* $G$ is trained on a source distribution different from the target distribution of the signal.

For example, in a task of recovering human faces, a 3-step DDIM generator trained on face images is weak along the generator-quality axis: it is matched to the target domain, but its unconditional samples are low-fidelity (see the top-left image in Figure 1). A 3-step DDIM generator trained on bedroom images is weak along both axes: it is low-quality and also mismatched to the target distribution (see the bottom-left image in Figure 1). Similarly, a full bedroom-trained diffusion model used to recover faces would be weak along the domain-match axis, even if its unconditional bedroom samples are high quality.

Our main experiments focus on these few-step matched and few-step mismatched settings with initial-noise optimization; we further separate the effect of domain mismatch from the choice of inverse solver in Appendix C.9.

**Diffusion inverse problem solvers:** Given a diffusion prior $\pi_{\mathrm{prior}}$ and measurement model $y = \mathcal{A}(x) + \epsilon$, the goal is to characterize the posterior $\pi(x \mid y)$. In practice, one can approximately sample from $\pi(x \mid y)$ or compute the maximum a posteriori (MAP) estimate $\arg\max_x \pi(x \mid y)$. Most existing solvers use the *full diffusion model* as the prior and follow a similar recipe: at each step, they adjust the backward update to incorporate information from the observation $y$ and the measurement model. This adjustment has been implemented in many ways, such as guidance, variable splitting, and sequential Monte Carlo; see Zheng et al. (2025) for a summary. Among them, diffusion posterior sampling (DPS) (Chung et al., 2023) is one of the most popular solvers, so we take it as our baseline. DPS has also been interpreted as approximate MAP optimization

(Xu et al., 2025).

**Initial noise optimization:** Another line of work treats the generative model $G$ as a black-box generator and solves inverse problems by optimizing the latent input. Given an observation $y$, it computes

$$\arg\min_z L\left(\mathcal{A}(G(z)), y\right) + \lambda \mathcal{R}(G(z)), \qquad (1)$$

where $L$ measures the mismatch between the simulated measurement $\mathcal{A}(G(z))$ and $y$, and $\mathcal{R}$ is a regularizer.

This latent-optimization view is conceptually simple and dates back to work on GANs and VAEs (Bora et al., 2017). However, when $G$ is a full diffusion model, optimizing (1) requires backpropagating through tens to hundreds of sampling steps, which is often too costly in memory and compute. As a result, prior work has to use a very weak generator $G$ (e.g., a 1–4 step DDIM sampler) (Wang et al., 2024; Chihaoui et al., 2024). Surprisingly, such inverse problem solvers based on weak generators can outperform full diffusion step-modification methods, which motivates our study.

Beyond inverse-problems, initial-noise techniques have recently attracted interest in other tasks for their potential to support inference-time scaling; see Ben-Hamu et al. (2024); Jia et al. (2026a); Ma et al. (2025); Zhou et al. (2025); Wan et al. (2025); Tang et al. (2025); Jia et al. (2026b).

**ADAMSPHERE and HOLDOUTTOPK early stopping:** Two factors, concentration and overfitting, affect the solution quality of (1). First, diffusion models are trained with $z \sim \mathbb{N}(0, I_d)$, whose mass in high dimension concentrates near the sphere $\|z\| \approx \sqrt{d}$ (Vershynin, 2018). Unconstrained optimization can push $z$ off this typical shell and degrade reconstruction quality. We therefore propose ADAMSPHERE, a modified ADAM optimizer that keeps $z$ on the sphere throughout. Second, it has been observed (Wang et al., 2024) that latent optimization can overfit measurement noise. We propose HOLDOUTTOPK early stopping. We hold out a subset of measurements (not used in optimization) and track its loss at each iteration. We then return the latest iterate among the best $K$ holdout losses. Our idea is inspired by statistical machine learning, where a validation set is used to approximate test performance and guide model selection. However, unlike the usual practice in machine learning of selecting the single lowest-validation-error candidate, we keep the top-$K$ and return the latest iterate among them. We find that $K > 1$ reduces sensitivity to noisy fluctuations while still selecting a near-best checkpoint that generalizes to the holdout set.

Since ADAMSPHERE already constrains the iterates to lie on the sphere, we optimize only the mean-squared error (MSE):

$$\arg\min_z \|\mathcal{A}(G(z)) - y\|^2. \qquad (2)$$

Section 3.2 shows that, under suitable assumptions, minimizing (2) is equivalent to steering the posterior toward the correct mode. The details of ADAMSPHERE and HOLD-OUTTOPK are given in Appendix B. Section 4.2 presents ablation studies demonstrating their effectiveness.

## 2.2. Main observations

Our main observation is that weak diffusion generators can still deliver strong performance on many practical inverse problems. Figure 1 shows representative examples. The left panel illustrates an inpainting task on a human face using two different weak priors, where the observed image (measurement) is obtained by masking a large fraction of pixels and adding Gaussian noise to the remaining pixels. The reference column shows the clean image, but it is never used by the reconstruction algorithm. Both rows use a 3-step DDIM prior: the prior in the top row (in the left panel) is trained on human face images, while the prior in the bottom row is trained on bedroom images. We reconstruct the noisy image using the initial-noise optimization procedure in Section 2.1, and show intermediate reconstructions at optimization iterations $1, 5, 10, 100$, and $1000$. In particular, `Iteration 1` corresponds to the unconditional 3-step DDIM sample, before any optimization is performed.

The priors in both rows are "weak." In the top row, the prior is in-distribution but low quality: the unconditional sample captures the rough outline of a face, but most colors and finer details are missing. The bottom row uses an even weaker, out-of-distribution prior: the unconditional sample is a low-quality sketch of a bedroom image. Nevertheless, as optimization proceeds, both priors quickly fit the measurement and converge to reconstructions that are both visually similar and accurate. The trajectory under the bedroom-trained prior is especially striking: early iterates look like a bedroom scene with straight edges and box-like shapes; but as optimization goes on, these features fade and the image gradually turns into a human face, with smoother contours and more detail. This firmly answers the question at the beginning of our paper: even a low-fidelity diffusion model trained on bedrooms can recover face images.

The right panel of Figure 1 shows a super-resolution experiment with similar conclusions. These examples are only a small sample of our study. In Section 4.1, we present larger-scale evaluations across multiple tasks, comparing weak priors against strong-prior baselines. Overall, the results suggest that weak diffusion priors can match strong-prior baselines in data-informative regimes.

## 3. Why weak priors can work

We now study why weak diffusion priors can still lead to accurate reconstructions. We identify two complementary

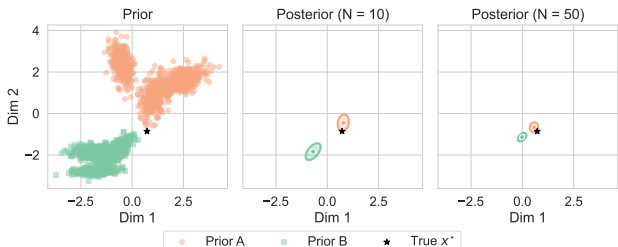

*Figure 2.* Posterior concentrates around $x^\star$ as $N$ grows.

mechanisms. First, when the measurement is sufficiently informative, the observation can dominate the prior: different priors can lead to similar posteriors because the posterior concentrates near the same measurement-consistent signal. Sections 3.1 and 3.2 formalize this effect through Bayesian-consistency arguments and a high-dimensional Gaussian-mixture surrogate. Second, weak priors are not arbitrary: even few-step or domain-mismatched natural-image priors can share local spatial statistics with stronger matched priors. Section 3.3 provides a local-correlation diagnostic supporting this second mechanism. After presenting these two mechanisms, we discuss when they break down in Section 3.4.

### 3.1. Posterior consistency under i.i.d. data

We first recall a standard phenomenon in Bayesian inference: under an i.i.d. model, sufficiently informative data can dominate the prior. Consider the linear inverse problem $y = Ax + \epsilon$, where $\epsilon$ is the standard Gaussian noise. In a synthetic experiment, we fix a ground-truth signal $x^\star$, draw a linear operator $A$, and generate i.i.d. observations $y_{1:N}$. We then place two well-separated Gaussian-mixture priors $\pi_A$ and $\pi_B$ on $x$ (Figure 2 left panel). Despite their large difference, as $N$ increases both posteriors $\pi_A(\cdot \mid y_{1:N})$ and $\pi_B(\cdot \mid y_{1:N})$ concentrate around $x^\star$, as shown in the right two panels of Figures 2.

This is an instance of *posterior consistency*. At a high level, the posterior is determined by both the prior and the data. As the data become increasingly informative (here, as the sample size grows), the likelihood term ultimately overwhelms the prior. Thus, even when the priors are very different, the resulting posteriors can still both concentrate on the same ground-truth signal. A classical result due to Schwartz (1965) makes this precise. For readability, assumptions and the exact convergence notion are deferred to the Appendix A.1. Recently, related posterior-contraction arguments have also been used to establish universal priors for empirical Bayes (Cannella et al., 2026).

**Theorem 3.1** (Posterior consistency, informal)**.** *Let $\{P_x\}_{x \in \mathcal{X}}$ be a statistical model and let $Y_1, Y_2, \ldots$ be i.i.d. from $P_{x^\star}$. Under standard regularity and identifiability as-*

*sumptions, and assuming the prior $\pi$ puts positive mass in every neighborhood of $x^\star$, the posterior $\pi(\cdot \mid y_{1:N})$ concentrates at $x^\star$ as $N \to \infty$.*

Theorem 3.1 shows that, under the i.i.d. model, the effect of the prior becomes negligible as the sample size grows. While insightful, the i.i.d. setting differs from image inverse problems, where measurements (pixels) are strongly dependent and we typically have only a single observation $y^\star$. The next section develops theory to study the posterior behavior in high-dimensional, single-observation regime.

### 3.2. Data dominance in high-dimensional inverse problems

We continue to study the linear inverse problem $y = Ax + \epsilon$, where $x \in \mathbb{R}^n$, $A \in \mathbb{R}^{m \times n}$, and $\epsilon \sim \mathbb{N}(0, \sigma^2 I_m)$. Throughout, we focus on the practically relevant regime where only a *single* observation $y^\star$ is available (e.g., one corrupted image), while the measurement dimension $m$ is large (e.g., $512 \times 512 \times 3$). Our goal is to understand when the posterior is *robust* to the choice of generative prior, so that substantially different priors can still yield similar reconstructions.

Throughout this section, we use $\varphi(\cdot, \mu, \Sigma)$ to denote the density of $\mathbb{N}(\mu, \Sigma)$.

**A tractable model for generative priors.** Modern generative priors such as $k$-step DDIM do not admit a tractable density. However, in practice their samples tend to lie near the data manifold, in the sense that generated images are often close to plausible clean-image prototypes. Motivated by this, a common surrogate is to model the prior as a Gaussian mixture. Finite Gaussian mixture models can approximate a broad class of distributions arbitrarily well (McLachlan & Peel, 2000). They are therefore widely adopted in studies of diffusion models, see Shah et al. (2023); Wu et al. (2024); Gatmiry et al. (2025); Liang et al. (2025). In particular, we assume the prior is an isotropic Gaussian mixture

$$\pi(x) = \sum_{j=1}^{M} w_j \varphi(x; \mu_j, \tau^2 I_n). \tag{3}$$

Here, $\{\mu_j\}_{j=1}^{M}$ are the component means, which can be interpreted as the universe of clean data. We interpret *different generative priors* as different collections of weights $\{w_j\}_{j=1}^{M}$. Assuming all components share a common variance $\tau^2$ keeps the core mathematics intact while keeping the argument clean. In the Appendix, we extend the model to allow component-specific (heterogeneous) variances.

**Posterior collapse to the best-scoring mode.** Given an observed $y^\star$, it is known that the posterior remains a Gaussian mixture. The posterior weight of component $j$ is proportional to $\tilde{w}_j := w_j \cdot \varphi(y^\star; A\mu_j, \Sigma)$, where $\Sigma := \sigma^2 I_m + \tau^2 A A^\top$. See Appendix A.2 for a proof.

For each component $j$, define the *score* as the squared measurement residual

$$s_j(y^\star) := \frac{1}{2} \|\Sigma^{-1/2}(y^\star - A\mu_j)\|_2^2. \tag{4}$$

Let $s_{(1)}(y^\star)$ and $s_{(2)}(y^\star)$ be the lowest and second-lowest scores. We define the *per-dimension score gap* as

$$\delta(y^\star) := \frac{1}{m}\Big(s_{(2)}(y^\star) - s_{(1)}(y^\star)\Big). \tag{5}$$

The following theorem states that, as $m$ grows, the posterior concentrates on the component with the smallest score.

**Theorem 3.2.** *With the notation above, assume:*

1. *(Bounded weight ratio) There exists $C > 1$ such that $1/C \le w_i/w_j \le C$ for all $(i, j)$.*

2. *($\delta_0$-identifiable) There is a unique minimizer $j^\star = \arg\min_j s_j(y^\star)$, and the score gap $\delta(y^\star) \ge \delta_0$ for a constant $\delta_0 > 0$.*

*Let $J$ be a categorical random variable on $\{1, 2, \ldots, M\}$ with $\mathbb{P}(J = j) := \tilde{w}_j / (\sum_{i=1}^{M} \tilde{w}_i)$, i.e., $J$ indexes the component selected in the Gaussian mixture posterior. Then*

- *The probability of not selecting the lowest score component satisfies*

$$\mathbb{P}(J \ne j^\star) \le CM \exp(-\delta_0 m).$$

- *The posterior concentrates onto the $j^\star$-th Gaussian component at an exponential rate in the measurement dimension $m$:*

$$\|\pi(\cdot \mid y^\star) - \mathbb{N}(m_{j^\star}(y^\star), \Sigma_{post})\|_{\mathsf{TV}} \le CM \exp(-\delta_0 m),$$

*where $m_{j^\star}(y^\star) = \mu_{j^\star} + \tau^2 A^\top \Sigma^{-1}(y^\star - A\mu_{j^\star})$, $\Sigma_{post} = \tau^2 I_n - \tau^4 A^\top \Sigma^{-1} A$, $\mathsf{TV}$ stands for total-variation distance.*

**Interpretation.** Theorem 3.2 shows that if the best-matching component is $\delta_0$-identifiable, then the posterior concentrates overwhelmingly on this component, with the remaining mass decaying exponentially in dimension $m$. A simple intuition for Assumption 2 is: in image inverse problems like random inpainting, once many pixels are observed, incorrect candidate images often disagree with the observation in many places (edges, textures, colors), so the best-matching candidate becomes clearly separated.

Notably, Theorem 3.2 is largely insensitive to the particular choice of generative prior: as long as the mixture weights are bounded within constant factors, the prior affects the bound only through constants. Therefore, in the data-informative regime (i.e., when Assumption 2 holds) and with high-dimensional observations (large $m$), very different priors can still yield similar reconstructions, with the posterior concentrating on the same mode.

**Score and MSE.** We remark that the score in (4) coincides (up to a constant factor) with the MSE in many practical inverse problems. This justifies our optimization procedure (2) used in Section 2.1: minimizing the MSE is often equivalent to minimizing the score.

For inpainting problems, the measurement operator $A = P_\Omega$ is a coordinate projection (i.e., it selects a set $\Omega$ of the observed pixels), so $AA^\top = I_m$ and the $\Sigma$ simplifies to $(\sigma^2 + \tau^2)I_m$. Therefore,

$$s_j(y^\star) = \frac{1}{2(\sigma^2 + \tau^2)} \sum_{i \in \Omega} \left(y^{\star,i} - \mu_{j,i}\right)^2,$$

which is exactly the MSE restricted to observed pixels.

For super-resolution, the measurement operator $A$ produces a low-resolution image $y \in \mathbb{R}^m$ from a high-resolution signal $x \in \mathbb{R}^n$ by averaging over non-overlapping downsampling blocks: each row of $A$ has $k$ nonzero entries with weights $1/k$, the row supports are disjoint, and $m \approx n/k$. Therefore, $AA^\top = k^{-1}I_m$, and the score simplifies to $s_j(y^\star) = (2k(\sigma^2 + \tau^2))^{-1}\|y^\star - A\mu_j\|_2^2$, which is again proportional to the MSE.

**Justifying the identifiable assumption.** We justify the $\delta_0$-identifiable assumption using both theory and empirical results. We evaluate three datasets: CelebA, Church, and Bedroom. For each image (with pixel values normalized to $(-1, 1)$), we generate an inpainting observation by masking 70% of pixels and adding Gaussian noise to the remaining pixels. For each resulting observation, we apply the same forward mask to every image in the dataset and compute the MSE on the observed pixels (which is proportional to the score). We then compute the per-dimension MSE gap for each image and report aggregated gap statistics in Table 1.

These results indicate a clear separation: the mean per-dimension gap is about 0.22–0.28. Interpreting this as a per-pixel squared difference, it corresponds to an average absolute difference of at least ($\sqrt{0.22} \approx 0.47$) on the observed entries; since each pixel lies in $[-1, 1]$ (range 2), this is a substantial mismatch. Moreover, the gap stays bounded away from zero even in the worst case: the minimum per-dimension MSE gap over all images is at least 0.09 (average absolute difference of at least 0.3). Thus, these suggest that the best-matching candidate is typically well separated from the runner-up. In Appendix A.4, we provide a theo-

*Table 1.* Random inpainting score gap. For each target image, we compute the per-dimension gap between the smallest and second-smallest observed-pixel MSE among all candidate images. All reported statistics are computed over 100 target images.

| Dataset | Mean (Std) | Min |
|---|---|---|
| CelebA | 0.22 (0.06) | 0.11 |
| Church | 0.28 (0.06) | 0.15 |
| Bedroom | 0.27 (0.07) | 0.09 |

retical justification for Assumption 2 using concentration inequalities.

The same bound also points to possible failure modes: posterior concentration can weaken when the score gap $\delta_0$ is small or when the effective measurement dimension $m$ is small. We return to these cases in Section 3.4, after introducing the local-structure view.

### 3.3. Shared local structure in weak and stronger priors

The posterior-concentration result above explains one side of the phenomenon: informative measurements can reduce sensitivity to the prior. It leaves open a second question: are 'weak priors' truly weak? That is, even when their samples are low-fidelity or mismatched, do they still preserve useful structural information that helps reconstruction?

We answer this question through a simple diagnostic: the spatial autocorrelation profile of unconditional samples. It measures how quickly pixel values decorrelate as spatial distance increases, and therefore captures short-range local structural information in the generated images.

We compare diffusion priors trained on CelebA-HQ and LSUN Bedroom, sampled with either 3 or 20 DDIM steps. For each model–sampler pair, we compute the spatial autocorrelation profile of their samples; the full definition and implementation details are given in Appendix C.8.

Table 2 reports representative lags. The main pattern is that the autocorrelation profiles are highly similar across both axes of variation: sampler strength and training domain. In particular, changing from 20 DDIM steps to 3 DDIM steps greatly reduces unconditional sample fidelity, and changing from CelebA-HQ to LSUN Bedroom changes the semantic content of the prior. Nevertheless, all four profiles show a similar decay with distance. For example, the Pearson correlation between the 3-step Bedroom profile and the 20-step CelebA-HQ profile over lags $1, \ldots, 32$ is 0.9987. This suggests that weak priors can lose sample fidelity or domain match while still preserving similar short-range local structural information.

| Lag | CelebA 3 | CelebA 20 | Bedroom 3 | Bedroom 20 |
|---|---|---|---|---|
| 0 | 1.0000 | 1.0000 | 1.0000 | 1.0000 |
| 1 | 0.9558 | 0.9814 | 0.9645 | 0.9615 |
| 2 | 0.9357 | 0.9594 | 0.9370 | 0.9260 |
| 4 | 0.8866 | 0.9100 | 0.8786 | 0.8573 |
| 8 | 0.7767 | 0.8108 | 0.7637 | 0.7437 |
| 16 | 0.5595 | 0.6261 | 0.5632 | 0.5618 |
| 32 | 0.2666 | 0.3481 | 0.3052 | 0.3207 |

*Table 2.* Spatial autocorrelation of samples from diffusion priors trained on CelebA-HQ and LSUN Bedroom, sampled with 3 or 20 DDIM steps. Values are averaged over 100 samples, RGB channels, and all directions at each rounded pixel distance.

**Mechanistic explanation:** Together with Theorem 3.2, this gives a complementary explanation for why weak priors can work. Informative measurements provide local anchors by fixing or strongly constraining many pixels, while the prior helps propagate these constraints to nearby unobserved or corrupted pixels. Because weak and stronger natural-image priors share similar short-range local correlations, even a weak prior can provide useful local structural information for this propagation. Thus, weak priors can be poor unconditional generators while still serving as useful conditional priors when the measurement provides enough anchors.

### 3.4. When weak priors fail

The two mechanisms above also clarify when weak diffusion priors fail. First, data dominance can break down when the observation does not identify a unique measurement-consistent signal. In large box inpainting, for example, the operator $A$ removes a contiguous region. Many images can match the observed pixels while differing inside the missing box, so the per-dimension score gap in Assumption 2 can be very small. The posterior then need not concentrate on a single mode, and reconstruction becomes prior-dominated. Second, local structural information is not enough when the task requires global semantic generation. In box inpainting, the missing region must be filled using information that is largely absent from the measurement. A mismatched prior may then propagate local statistics but still generate semantic content inconsistent with the target image.

A similar issue appears when the effective measurement dimension is small. The bound in Theorem 3.2 contains the factor $M \exp(-\delta_0 m)$: it grows linearly with the number of candidate modes $M$ and decays exponentially in the number of observed measurements $m$. For 70% random inpainting on a $256 \times 256$ RGB image, $m \approx 256 \times 256 \times 30\% \times 3 \approx 60{,}000$, so the exponential term can dominate. For $16\times$ super-resolution, however, $m$ can be as low as $16 \times 16 \times 3 = 768$. In such low-information regimes, the measurement provides too few anchors, so reconstruction becomes much more sensitive to the strength and domain match of the prior. We empirically validate these failure modes in Section 4.3.

## 4. Experiments

We evaluate our method on four tasks: inpainting, Gaussian deblurring, super-resolution, and nonlinear deblurring with additive Gaussian noise. For each subsection, we report the subset of tasks most relevant to the setting under study, while full results across are provided in Appendix C. Additional comparisons with DAPS (Zhang et al., 2025) and DiffPIR (Zhu et al., 2023), Deep Random Projector, and scientific inverse-problems are provided in Appendices C.3, C.4, and C.5. We study sensitivity to measurement noise in

Appendix C.10. We evaluate each method using standard metrics, e.g. PSNR, SSIM, and LPIPS (Zhang et al., 2018).

Unless otherwise stated, all experiments are conducted using publicly available pretrained diffusion checkpoints for the Bedroom (LSUN-Bedroom (Yu et al., 2015)), Church (LSUN-Church (Yu et al., 2015)), and Human face (CelebA-HQ (Liu et al., 2015)) datasets. For each setting, we randomly select 100 test images for evaluation. For the ablation studies, we use a reduced set of 50 images. Detailed experiment setup is provided in Appendix C.12. Appendix D provides additional visualizations.

**Compute note.** Our goal is to study robustness to weak priors rather than to provide a compute-matched benchmark. We therefore report each method under its default settings; compute-matched comparisons are given in Appendix C.11.

### 4.1. Cross-Domain Inverse Problem Solving

Recall that weak priors can arise either from *(1) using a few-step sampler* or *(2) training on a mismatched dataset.* We therefore design the following cross-domain experiments to evaluate performance under both types of weak priors.

We consider four restoration tasks: inpainting, Gaussian deblurring, $4\times$ super-resolution, and nonlinear deblurring for bedroom, church, and human face images. For each task–dataset pair (e.g., human face inpainting), we benchmark reconstruction under a strong prior using the DPS algorithm (Chung et al., 2023), which uses a 1000-step DDPM trained on the same distribution as the target image. We then compare against two weak-prior settings: a 3-step DDIM prior trained on matched dataset, and a 3-step DDIM prior trained on mismatched dataset (e.g., LSUN-Church or LSUN-Bedroom). For each weak prior, we solve the corresponding task using initial-noise optimization with our optimizer and HOLDOUTTOPK early-stopping strategy, as described in Section 2.1. We choose a 3-step DDIM prior following the recommendation of Wang et al. (2024), which includes an ablation study over the number of DDIM steps.

Table 3 reports results for inpainting and $4\times$super-resolution, with the remaining tasks deferred to Appendix C. The results suggest weak prior can achieve accurate reconstructions. With initial-noise optimization, the few-step, in-domain prior overall outperforms the DPS baseline (which uses an in-domain, 1000-step prior) across all metrics. For example, our method achieves a PSNR gain of 0.88–1.80 dB on inpainting, and an even larger gain on super-resolution. These results are also consistent with Wang et al. (2024) and Chihaoui et al. (2024). Somewhat surprisingly, even with the weakest prior (out-of-domain and few-step generator), performance can still match or exceed DPS. For example, using a bedroom-pretrained model for CelebA inpainting, our method achieves PSNR 32.76, compared to DPS's 31.98.

Thus, these results show that even few-step, out-of-domain diffusion generators can serve as effective priors. [1]

We also study how domain mismatch affects reconstruction quality. Specifically, we compare our method using a few-step generator under matched versus mismatched source–target domains. As shown in Table 3, the in-domain generator outperforms the out-of-domain generator. The gap is sometimes moderate (e.g., CelebA inpainting) and sometimes small (e.g., Bedroom inpainting). Meanwhile, using a bedroom-trained model versus a church-trained model often yields similar performance, especially compared with CelebA-trained model. This matches the intuition that bedrooms and churches are more similar to each other than to human faces. These comparisons suggest that domain mismatch can affect performance, but the effect is often modest, smaller when the source and target domains are visually closer and larger when they are farther apart.

### 4.2. Comparison with Optimization-Based Methods and Ablations

Next, we compare our method with the state-of-the-art optimization-based approach DMPlug (Wang et al., 2024). Both methods optimize the initial noise, but with different optimizers and early-stopping strategies. To isolate the effect of algorithmic design, we use the same in-domain 3-step DDIM prior (as suggested in (Wang et al., 2024)) and run both algorithms from the same randomly sampled initial noise, for the same number of iterations. Results for random inpainting and nonlinear deblurring are summarized in Table 4. Other tasks results are provided in Appendix C. Our PSNR exceeds DMPlug in most cases, with gains ranging from 0.15 to 1 dB; in the remaining cases, the difference is below 0.1 dB. Our LPIPS is consistently lower than DM-Plug's, with improvement up to 30%.

We also include a small ablation study of our new optimizer, ADAMSPHERE, and our early-stopping strategy, HOLDOUT-TOPK; the results are reported in Table 5. They indicate that ADAMSPHERE matches the optimization performance of standard Adam, while explicitly keeping the noise near the typical Gaussian shell, which has been found important for high-quality generation (Yang et al., 2024; Mannering et al., 2025). Meanwhile, HOLDOUTTOPK consistently improves over the variance-based early-stopping baseline used in Wang et al. (2024). Thus, our strategy offers a simple, effective alternative for reducing overfitting.

---

[1]Although our method often outperforms DPS on these metrics, we do not claim that a weak prior is generally better than a strong prior. DPS is a widely used strong-prior baseline, but it is not necessarily the best inverse-problem solver for every task or dataset. Still, consistent gains over DPS show that weak, even out-of-domain priors can be practically effective when the measurements are sufficiently informative.

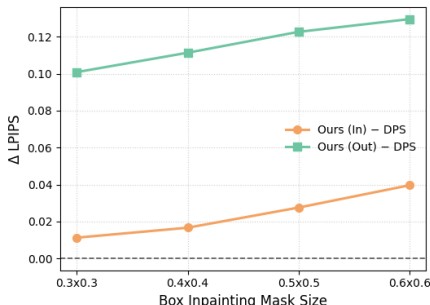

*Figure 4.* Difference in LPIPS between our method and the DPS baseline for different mask sizes. The dashed horizontal line denotes equal performance with DPS.

### 4.3. Failure Modes

We examine the limitations of weak priors. To study this, we consider *box inpainting* and *large-scale super-resolution*, which are closer to generation than reconstruction. For example, box inpainting requires filling an entire region.

In both experiments, the goal is to recover church images. As before, DPS uses a full 1000-step DDPM prior trained on Church, while our method uses a 3-step DDIM prior trained on Church (in-domain) and on CelebA (out-of-domain), respectively. For box inpainting, we mask a contiguous square region whose size increases across trials, and compare our method against the DPS baseline. Results are plotted in Figure 4 and recorded in Table 12 in Appendix C. We observe a clear failure mode: as the masked fraction increases, our method degrades and performs worse than DPS, with the performance gap growing as masking becomes more severe. In particular, the out-of-domain prior fails completely, as reflected in both the metrics and Figure 3 top row. The filled region has a face-like appearance that is clearly inconsistent with the church content. Similar observations appear for super-resolution. As the super-resolution factor increases, performance degrades sharply and essentially all methods fail. However, as shown in Figure 3 bottom row, DPS still generates higher-quality samples due to its stronger prior. In contrast, weak priors produce blurry, inconsistent samples.

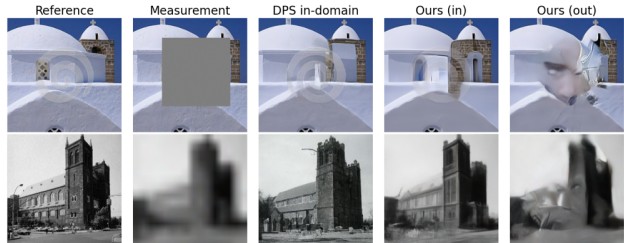

*Figure 3.* Visual comparison on box inpainting and super-resolution tasks. **Top row**: box inpainting with a $0.6 \times 0.6$ mask. **Bottom row**: $16\times$ super-resolution.

*Table 3.* Cross-Domain results for inpainting and super-resolution. The "Model" column indicates the source domain of the pretrained diffusion model. The "CelebA," "Bedroom," and "Church" columns indicate the target domains of the images being reconstructed. DPS always uses an in-domain model, i.e., the source and target domains coincide.

| Task | Method | Model | CelebA | | | Bedroom | | | Church | | |
|------|--------|-------|--------|--------|--------|---------|--------|--------|--------|--------|--------|
| | | | PSNR ↑ | LPIPS ↓ | SSIM ↑ | PSNR ↑ | LPIPS ↓ | SSIM ↑ | PSNR ↑ | LPIPS ↓ | SSIM ↑ |
| **Inpainting** | DPS | In-domain | 31.98 | **0.14** | 0.88 | 27.97 | 0.23 | 0.82 | 24.15 | 0.25 | 0.73 |
| | Ours | CelebA | **33.78** | 0.15 | **0.91** | 27.78 | 0.27 | 0.83 | 23.56 | 0.36 | 0.72 |
| | | Bedroom | 32.76 | 0.17 | 0.90 | **28.88** | **0.20** | **0.87** | 24.22 | 0.28 | 0.75 |
| | | Church | 32.62 | 0.16 | 0.90 | 28.66 | 0.21 | 0.86 | **24.93** | **0.23** | **0.77** |
| **Super-Res** | DPS | CelebA | 26.82 | 0.22 | 0.74 | 22.95 | 0.39 | 0.64 | 20.28 | 0.36 | 0.52 |
| | Ours | CelebA | **31.27** | **0.18** | **0.86** | 25.88 | 0.35 | 0.74 | 22.68 | 0.41 | 0.63 |
| | | Bedroom | 30.34 | 0.22 | 0.84 | **26.59** | **0.25** | **0.78** | 22.86 | 0.33 | 0.65 |
| | | Church | 30.10 | 0.22 | 0.84 | 26.30 | 0.27 | 0.77 | **23.09** | **0.27** | **0.67** |

*Table 4.* Comparison with DMPlug under in-domain priors for random inpainting and nonlinear deblurring task.

| Task | Dataset | Ours | | | DMplug | | |
|------|---------|------|--------|--------|--------|--------|--------|
| | | PSNR ↑ | LPIPS ↓ | SSIM ↑ | PSNR ↑ | LPIPS ↓ | SSIM ↑ |
| **Inpainting** | CelebA | **33.784** | **0.147** | **0.915** | 32.778 | 0.189 | 0.892 |
| | Church | 24.927 | **0.234** | **0.771** | **24.957** | 0.246 | 0.771 |
| | Bedroom | **28.883** | **0.201** | **0.867** | 28.271 | 0.255 | 0.836 |
| **Nonlinear** | CelebA | **25.21** | **0.27** | **0.74** | 24.85 | 0.28 | 0.73 |
| | Church | **20.88** | **0.39** | **0.56** | 20.48 | 0.42 | 0.53 |
| | Bedroom | **22.67** | **0.39** | **0.64** | 22.21 | 0.41 | 0.63 |

| Task | Dataset | (A) | (B) | (C) | (D) |
|------|---------|-----|-----|-----|-----|
| Gaussian | Church | 22.034 | 22.394 | 22.048 | **22.404** |
| | Bedroom | 24.825 | 25.595 | 24.854 | **25.596** |
| Inpainting | Church | 24.724 | 24.843 | 24.743 | **24.844** |
| | Bedroom | 28.669 | 29.156 | 28.559 | **29.162** |

*Table 5.* PSNR for Gaussian deblurring and inpainting. (A) Adam + variance, (B) Adam + HOLDOUTTOPK, (C) ADAMSPHERE + variance, (D) ADAMSPHERE + HOLDOUTTOPK.

consistent signal. The correlation diagnostics show that weak and mismatched natural-image priors can still share local spatial structure with stronger matched priors. Together, these results explain why weak priors can work in data-informative regimes and why they fail when measurements provide too few anchors.

Practically, our results suggest a simple guide: weak priors are useful defaults when measurements provide many local anchors, as in random inpainting, deblurring, and moderate super-resolution. In these settings, reconstruction is driven largely by the observation, while the prior helps propagate local constraints through shared spatial structure. Stronger matched priors become more important when the observation leaves a large null space, has high noise, or requires global semantic completion, as in large box inpainting or high-factor super-resolution.

Looking forward, this phenomenon calls for further study of algorithms tailored to weak priors, especially few-step priors. Despite the strong performance of initial-noise optimization, existing work remains limited and largely centers on this single approach. Potential algorithmic improvements include hybrid methods that combine measurement injection with noise optimization, and more effective sampling procedures. On the theoretical side, further work is needed to pin down sharp thresholds that determine when measurements are informative enough for weak priors to be effective.

## 4.4. Latent Diffusion Applications on ImageNet

We test few-step latent diffusion priors using Stable Diffusion 2.1 (Rombach et al., 2022) and DiT (Peebles & Xie, 2023). These results, reported in Table 13 in Appendix C.7, show the same qualitative pattern: few-step latent priors still give accurate reconstructions in data-informative settings.

## 5. Conclusion, Practical Guide, and Discussion

This work studies when inverse-problem performance is robust to the choice of prior. We show that weak priors, arising either from low-fidelity generators or from training–reconstruction domain mismatch, can still perform well across many inverse problems. To explain this phenomenon, we combine Bayesian-consistency theory with local-correlation diagnostics. The theory shows that sufficiently informative measurements can reduce sensitivity to the prior by concentrating the posterior near a measurement-

## Acknowledgement

The authors thank Haochen Ji and Qian Qin for helpful discussions. The authors thank the four anonymous reviewers for their insightful suggestions during the rebuttal process. Guanyang Wang and Jing Jia acknowledge support from the National Science Foundation through grant DMS–2210849 and an Adobe Data Science Research Award. Liyue Shen acknowledges funding support by NSF (National Science Foundation) via grants IIS-2435746, Defense Advanced Research Projects Agency (DARPA) under Contract No. HR00112520042, as well as the University of Michigan MIDAS PODS Grant Award.

## Impact Statement

This work aims to improve the understanding of diffusion-model priors for inverse problems. It may help reduce the need for expensive or domain-specific generative models when measurements are sufficiently informative. At the same time, our results show that weak or mismatched priors can fail in low-information regimes, such as large box inpainting or high-factor super-resolution, where reconstructions may contain plausible but incorrect structure. In high-stakes settings such as medical or scientific imaging, such reconstructions should therefore be used only with appropriate validation, uncertainty checks, and domain expertise.

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

# A. Additional theoretical results and proofs

## A.1. Posterior consistency under i.i.d. data

The following theorem gives posterior consistency for i.i.d. data. It goes back to Schwartz (1965); see also Chapter 6 of Ghosal & Van der Vaart (2017) for extensions.

**Theorem A.1** (Posterior consistency). *Let $(\mathcal{X}, d)$ be a metric space and let $\{P_x\}_{x \in \mathcal{X}}$ be a dominated i.i.d. model: there exists a $\sigma$-finite measure $\mu$ such that $P_x \ll \mu$ with density $p_x = \frac{dP_x}{d\mu}$ for every $x \in \mathcal{X}$. Let $Y_1, Y_2, \ldots$ be i.i.d. from $P_{x^\star}$, and let $\pi$ be a prior on $\mathcal{X}$. Write $Y_{1:n} = (Y_1, \ldots, Y_n)$ and*

$$\pi(A \mid Y_{1:n}) = \frac{\int_A \prod_{i=1}^n p_x(Y_i)\, \pi(dx)}{\int_{\mathcal{X}} \prod_{i=1}^n p_x(Y_i)\, \pi(dx)}.$$

*Assume:*

*(i) **KL support at the truth:** for every $\varepsilon > 0$,*

$$\pi(\{x \in \mathcal{X} : \mathsf{KL}(P_{x^\star}, P_x) < \varepsilon\}) > 0, \qquad \mathsf{KL}(P_{x^\star}, P_x) := \int \log\left(\frac{p_{x^\star}}{p_x}\right) dP_{x^\star},$$

*where $\mathsf{KL}$ stands for the Kullback-Leibler (KL) divergence.*

*(ii) **Existence of uniformly consistent tests:** for every $\delta > 0$ there exists a sequence of tests $\phi_n = \phi_n(Y_{1:n}) \in [0, 1]$ such that*

$$\mathbb{E}_{x^\star}[\phi_n] \to 0 \quad and \quad \sup_{x:\, d(x, x^\star) > \delta} \mathbb{E}_x[1 - \phi_n] \to 0 \qquad as\ n \to \infty.$$

*Then the posterior is (strongly) consistent at $x^\star$: for every $\delta > 0$,*

$$\pi(\{x \in \mathcal{X} : d(x, x^\star) > \delta\} \mid Y_{1:n}) \to 0 \quad almost\ surely\ under\ P_{x^\star}^\infty.$$

## A.2. Close-formula for Gaussian mixture posterior

**Proposition A.2.** *Given the measurement model $y = Ax + \epsilon$, where $\epsilon \sim \mathbb{N}(0, \sigma^2 I_m)$, and a Gaussian mixture prior*

$$\pi(x) = \sum_{j=1}^M w_j\, \varphi(x; \mu_j, \Sigma_j), \qquad w_j \geq 0,\ \sum_{j=1}^M w_j = 1,$$

*where each $\Sigma_j \in \mathbb{R}^{n \times n}$ is symmetric positive definite. Then the posterior $\pi(x \mid y)$ is also a Gaussian mixture:*

$$\pi(x \mid y) = \sum_{j=1}^M \widetilde{w}_j(y)\, \varphi(x; m_j(y), C_j),$$

*where,*

$$C_j := \left(\Sigma_j^{-1} + \tfrac{1}{\sigma^2} A^\top A\right)^{-1},$$

$$m_j(y) := C_j\left(\Sigma_j^{-1} \mu_j + \tfrac{1}{\sigma^2} A^\top y\right).$$

*The updated weights satisfy*

$$\widetilde{w}_j(y) = \frac{w_j\, \varphi(y; A\mu_j,\ \sigma^2 I_m + A\Sigma_j A^\top)}{\sum_{\ell=1}^M w_\ell\, \varphi(y; A\mu_\ell,\ \sigma^2 I_m + A\Sigma_\ell A^\top)}.$$

*Proof.* The posterior density

$$\pi(x \mid y) = \frac{p(y \mid x)\, \pi(x)}{\int p(y \mid x)\, \pi(x)\, dx} = \frac{\sum_{j=1}^M w_j\, \varphi(y; Ax, \sigma^2 I_m)\, \varphi(x; \mu_j, \Sigma_j)}{\sum_{j=1}^M w_j \int \varphi(y; Ax, \sigma^2 I_m)\, \varphi(x; \mu_j, \Sigma_j)\, dx}.$$

Thus it suffices to analyze, for each fixed $j$, the product

$$\varphi(y; Ax, \sigma^2 I_m) \, \varphi(x; \mu_j, \Sigma_j).$$

Fix $j$. Consider the hierarchical model

$$X \sim \mathbb{N}(\mu_j, \Sigma_j), \qquad Y \mid X = x \sim \mathbb{N}(Ax, \sigma^2 I_m).$$

It is known that $(X, Y)$ is jointly Gaussian with mean $(\mu_j, \, A\mu_j)$ and covariance

$$\begin{pmatrix} \Sigma_j & \Sigma_j A^\top \\ A\Sigma_j & A\Sigma_j A^\top + \sigma^2 I_m \end{pmatrix}.$$

In particular, the marginal distribution of $Y$ under component $j$ is

$$p_j(y) := \int \varphi(y; Ax, \sigma^2 I_m) \, \varphi(x; \mu_j, \Sigma_j) \, dx = \varphi\big(y; A\mu_j, \ \sigma^2 I_m + A\Sigma_j A^\top\big).$$

Also, the conditional distribution of $X$ given $Y = y$ under component $j$ is Gaussian with covariance

$$\widetilde{C}_j = \Sigma_j - \Sigma_j A^\top \big(\sigma^2 I_m + A\Sigma_j A^\top\big)^{-1} A\Sigma_j$$

and mean

$$\widetilde{m}_j(y) = \mu_j + \Sigma_j A^\top \big(\sigma^2 I_m + A\Sigma_j A^\top\big)^{-1}(y - A\mu_j).$$

Using the Woodbury identity, one can check that

$$\widetilde{C}_j = \Big(\Sigma_j^{-1} + \tfrac{1}{\sigma^2} A^\top A\Big)^{-1} =: C_j,$$

and substituting this into the expression for $\widetilde{m}_j(y)$ gives the equivalent form

$$\widetilde{m}_j(y) = C_j \Big(\Sigma_j^{-1}\mu_j + \tfrac{1}{\sigma^2} A^\top y\Big) =: m_j(y).$$

Therefore, for each $j$ we have the factorization of the joint density

$$\varphi(y; Ax, \sigma^2 I_m) \, \varphi(x; \mu_j, \Sigma_j) = p_j(y) \, \varphi\big(x; m_j(y), C_j\big),$$

where $p_j(y) = \varphi\big(y; A\mu_j, \sigma^2 I_m + A\Sigma_j A^\top\big)$.

Plugging this into Bayes' rule yields

$$\pi(x \mid y) = \frac{\sum_{j=1}^M w_j \, p_j(y) \, \varphi(x; m_j(y), C_j)}{\sum_{\ell=1}^M w_\ell \, p_\ell(y)} = \sum_{j=1}^M \widetilde{w}_j(y) \, \varphi(x; m_j(y), C_j),$$

with

$$\widetilde{w}_j(y) = \frac{w_j \, p_j(y)}{\sum_{\ell=1}^M w_\ell \, p_\ell(y)} = \frac{w_j \, \varphi\big(y; A\mu_j, \ \sigma^2 I_m + A\Sigma_j A^\top\big)}{\sum_{\ell=1}^M w_\ell \, \varphi\big(y; A\mu_\ell, \ \sigma^2 I_m + A\Sigma_\ell A^\top\big)}.$$

This is exactly the stated Gaussian-mixture posterior with the given component means, covariances, and updated weights.

$\square$

### A.3. Posterior collapse under heterogeneous variances

Here we prove a more general version of Theorem 3.2 under a heterogeneous-variance assumption.

**Setup:**   Consider the measurement model

$$y = Ax + \epsilon, \qquad \epsilon \sim \mathbb{N}(0, \sigma^2 I_m),$$

and the Gaussian mixture

$$\pi(x) = \sum_{j=1}^{M} w_j \, \varphi\big(x; \mu_j, \tau_j^2 I_n\big), \qquad w_j \geq 0, \sum_{j=1}^{M} w_j = 1.$$

We note that this setting is strictly more general than the equal-variance setting in our main text (3), since each component is allowed to have its own covariance scale $\tau_j^2 I_n$. It reduces to the homogeneous case when $\tau_1 = \tau_2 = \cdots = \tau_M = \tau$.

We prove the following theorem:

**Theorem A.3** (Posterior collapse to the best *selection-score* mode)**.** *Consider the measurement model and Gaussian mixture prior as above. Let*

$$\Sigma_j := \sigma^2 I_m + \tau_j^2 A A^\top.$$

*Define the* selection score

$$\ell_j(y^\star) := s_j(y^\star) + \frac{1}{2} \log \det(\Sigma_j), \qquad s_j(y^\star) := \frac{1}{2} \big\| \Sigma_j^{-1/2}(y^\star - A\mu_j) \big\|_2^2. \qquad (6)$$

*Let $\ell_{(1)}(y^\star)$ and $\ell_{(2)}(y^\star)$ be the smallest and second-smallest values among $\{\ell_j(y^\star)\}_{j=1}^M$, and define the per-dimension selection-score gap*

$$\delta_\ell(y^\star) := \frac{1}{m}\Big(\ell_{(2)}(y^\star) - \ell_{(1)}(y^\star)\Big). \qquad (7)$$

*Assume:*

1. *(Bounded weight ratio) There exists $C > 1$ such that $1/C \leq w_i/w_j \leq C$ for all $i, j$.*

2. *($\delta_0$-identifiable in selection score) There is a unique minimizer*

$$j^\star = \arg \min_{1 \leq j \leq M} \ell_j(y^\star),$$

   *and $\delta_\ell(y^\star) \geq \delta_0$ for some constant $\delta_0 > 0$.*

*Let J be the posterior component index, i.e.*

$$\Pr(J = j) = \widetilde{w}_j(y^\star) := \frac{w_j \, \varphi\big(y^\star; A\mu_j, \Sigma_j\big)}{\sum_{i=1}^{M} w_i \, \varphi\big(y^\star; A\mu_i, \Sigma_i\big)}.$$

*Then*

- *The probability of* not *selecting the best selection-score component satisfies*

$$\Pr(J \neq j^\star) \leq CM \, e^{-m\delta_0}.$$

- *The posterior concentrates onto the $j^\star$-th Gaussian component at an exponential rate:*

$$\big\| \pi(\cdot \mid y^\star) - \mathbb{N}(m_{j^\star}(y^\star), \Sigma_{\text{post},j^\star}) \big\|_{\text{TV}} \leq CM \, e^{-m\delta_0},$$

   *where*

$$\Sigma_{\text{post},j} = \Big(\tau_j^{-2} I_n + \frac{1}{\sigma^2} A^\top A\Big)^{-1}, \qquad m_j(y^\star) = \Sigma_{\text{post},j}\Big(\tau_j^{-2}\mu_j + \frac{1}{\sigma^2} A^\top y^\star\Big).$$

*Remark* A.4.  Theorem A.3 reduces to Theorem 3.2 when $\tau_1 = \ldots = \tau_M = \tau$.

*Proof.* By Proposition A.2, the posterior weights satisfy

$$\widetilde{w}_j(y^\star) \;\propto\; w_j\,\varphi\big(y^\star; A\mu_j, \Sigma_j\big), \qquad \Sigma_j = \sigma^2 I_m + \tau_j^2 A A^\top.$$

Fix $j \neq j^\star$. Using the Gaussian density formula,

$$
\begin{aligned}
\frac{\widetilde{w}_j(y^\star)}{\widetilde{w}_{j^\star}(y^\star)}
&= \frac{w_j}{w_{j^\star}} \frac{\varphi\big(y^\star; A\mu_j, \Sigma_j\big)}{\varphi\big(y^\star; A\mu_{j^\star}, \Sigma_{j^\star}\big)} \\
&= \frac{w_j}{w_{j^\star}} \Big( \frac{\det(\Sigma_{j^\star})}{\det(\Sigma_j)} \Big)^{1/2} \exp\!\Big( -s_j(y^\star) + s_{j^\star}(y^\star) \Big) \\
&= \frac{w_j}{w_{j^\star}} \exp\!\Big( -\big[ s_j(y^\star) + \tfrac{1}{2}\log\det(\Sigma_j) \big] + \big[ s_{j^\star}(y^\star) + \tfrac{1}{2}\log\det(\Sigma_{j^\star}) \big] \Big) \\
&= \frac{w_j}{w_{j^\star}} \exp\!\Big( -\big( \ell_j(y^\star) - \ell_{j^\star}(y^\star) \big) \Big).
\end{aligned}
\tag{8}
$$

By the bounded weight-ratio assumption, $w_j/w_{j^\star} \leq C$ for all $j$, hence

$$\frac{\widetilde{w}_j(y^\star)}{\widetilde{w}_{j^\star}(y^\star)} \leq C \, \exp\!\Big( -\big( \ell_j(y^\star) - \ell_{j^\star}(y^\star) \big) \Big). \tag{9}$$

Next, note that

$$\Pr(J \neq j^\star) = \sum_{j \neq j^\star} \widetilde{w}_j(y^\star) = \widetilde{w}_{j^\star}(y^\star) \sum_{j \neq j^\star} \frac{\widetilde{w}_j(y^\star)}{\widetilde{w}_{j^\star}(y^\star)} \leq \sum_{j \neq j^\star} \frac{\widetilde{w}_j(y^\star)}{\widetilde{w}_{j^\star}(y^\star)},$$

since $\widetilde{w}_{j^\star}(y^\star) \leq 1$. Combining with (9) gives

$$\Pr(J \neq j^\star) \leq C \sum_{j \neq j^\star} \exp\!\Big( -\big( \ell_j(y^\star) - \ell_{j^\star}(y^\star) \big) \Big).$$

By definition of $\ell_{(2)}$ and $\ell_{(1)} = \ell_{j^\star}$, we have $\ell_j(y^\star) - \ell_{j^\star}(y^\star) \geq \ell_{(2)}(y^\star) - \ell_{(1)}(y^\star)$ for all $j \neq j^\star$. Therefore,

$$\Pr(J \neq j^\star) \leq C(M-1) \exp\!\Big( -\big( \ell_{(2)}(y^\star) - \ell_{(1)}(y^\star) \big) \Big) \leq CM \exp\!\big( -m\delta_\ell(y^\star) \big) \leq CM \exp(-m\delta_0),$$

which proves the first claim.

For the total-variation bound, write the posterior mixture as

$$\pi(\cdot \mid y^\star) = \widetilde{w}_{j^\star}(y^\star)\, \mathbb{N}(m_{j^\star}(y^\star), \Sigma_{\mathrm{post},j^\star}) + \sum_{j \neq j^\star} \widetilde{w}_j(y^\star)\, \mathbb{N}(m_j(y^\star), \Sigma_{\mathrm{post},j}).$$

Let $\mu^\star := \mathbb{N}(m_{j^\star}(y^\star), \Sigma_{\mathrm{post},j^\star})$. Then, using that $\|\nu_1 - \nu_2\|_{\mathsf{TV}} \leq 1$ for any probability measures $\nu_1, \nu_2$,

$$\big\| \pi(\cdot \mid y^\star) - \mu^\star \big\|_{\mathsf{TV}} = \Big\| \sum_{j \neq j^\star} \widetilde{w}_j(y^\star)\big( \mathbb{N}(m_j, \Sigma_{\mathrm{post},j}) - \mu^\star \big) \Big\|_{\mathsf{TV}} \leq \sum_{j \neq j^\star} \widetilde{w}_j(y^\star) \big\| \mathbb{N}(m_j, \Sigma_{\mathrm{post},j}) - \mu^\star \big\|_{\mathsf{TV}} \leq \sum_{j \neq j^\star} \widetilde{w}_j(y^\star).$$

Hence

$$\big\| \pi(\cdot \mid y^\star) - \mu^\star \big\|_{\mathsf{TV}} \leq \Pr(J \neq j^\star) \leq CM e^{-m\delta_0},$$

as claimed. $\square$

### A.4. Validating Assumption 2 in Theorem 3.2

We give a simple probabilistic argument showing that the $\delta_0$-identifiable condition in Assumption 2 holds with high probability for *random inpainting*, provided the candidate means are separated in full-image MSE.

**Setup.** Let $A = P_\Omega$ be a coordinate projection onto a uniformly random subset $\Omega \subset [n]$ with $|\Omega| = m$. Assume all pixel values are bounded: $|\mu_{j,i}| \leq 1$ for all $j, i$. Consider the Gaussian-mixture surrogate where the data are generated from a single component $j^\star$:

$$x^\star = \mu_{j^\star} + \xi, \qquad \xi \sim \mathbb{N}(0, \tau^2 I_n), \qquad y^\star = P_\Omega x^\star + \epsilon, \qquad \epsilon \sim \mathbb{N}(0, \sigma^2 I_m),$$

so that

$$y^\star = P_\Omega \mu_{j^\star} + \eta, \qquad \eta := P_\Omega \xi + \epsilon \sim \mathbb{N}\big(0, (\sigma^2 + \tau^2) I_m\big).$$

Recall that for inpainting,

$$s_j(y^\star) = \frac{1}{2(\sigma^2 + \tau^2)} \big\| y^\star - P_\Omega \mu_j \big\|_2^2.$$

Assume there exists $\Delta > 0$ such that for every $j \neq j^\star$,

$$\frac{1}{n} \big\| \mu_j - \mu_{j^\star} \big\|_2^2 \geq \Delta. \tag{10}$$

This is a full-image (population) MSE separation between the correct mean and all others, which is natural because visually different images typically differ across many pixels.

Then we show the following:

**Proposition A.5** (Identifiability for random inpainting). *Under the setup above and* (10), *define the per-dimension score gap*

$$\delta(y^\star) := \min_{j \neq j^\star} \frac{s_j(y^\star) - s_{j^\star}(y^\star)}{m}.$$

*Then there is a universal constant $c > 0$ such that*

$$\mathbb{P}\left( \delta(y^\star) \geq \frac{\Delta}{8(\sigma^2 + \tau^2)} \right) \geq 1 - 2(M-1)\exp\left(-\frac{1}{32} m\Delta^2\right) - 2(M-1)\exp\left(-\frac{m\Delta^2}{32(\sigma^2 + \tau^2)}\right).$$

*In other words, the score will be $\frac{\Delta}{8(\sigma^2+\tau^2)}$-identifiable with probability tending to 1 exponentially fast in $m$.*

*Proof.* We pick an arbitrary $j \neq j^\star$, write

$$y^\star - P_\Omega \mu_j = P_\Omega(\mu_{j^\star} - \mu_j) + \eta.$$

Therefore $y^\star - P_\Omega \mu_j \sim \mathbb{N}(P_\Omega(\mu_{j^\star} - \mu_j), (\sigma^2 + \tau^2)I_m)$. Let $v_j := P_\Omega(\mu_{j^\star} - \mu_j)$. Expanding and cancelling the common $\|\eta\|_2^2$ term gives

$$s_j(y^\star) - s_{j^\star}(y^\star) = \frac{1}{2(\sigma^2 + \tau^2)}\left(\|v_j + \eta\|_2^2 - \|\eta\|_2^2\right)$$

$$= \frac{1}{2(\sigma^2 + \tau^2)}\left(\|v_j\|_2^2 + 2\langle \eta, v_j \rangle\right).$$

Dividing by $m$, we have

$$\frac{s_j(y^\star) - s_{j^\star}(y^\star)}{m} = \frac{1}{2(\sigma^2 + \tau^2)}\left(\frac{1}{m}\|v_j\|_2^2 + \frac{2}{m}\langle \eta, v_j \rangle\right). \tag{11}$$

Therefore,

$$\mathbb{P}\left(\frac{s_j(y^\star) - s_{j^\star}(y^\star)}{m} \geq \frac{\Delta}{8(\sigma^2 + \tau^2)}\right) = \mathbb{P}\left(\left(\frac{1}{m}\|v_j\|_2^2 + \frac{2}{m}\langle \eta, v_j \rangle\right) \geq \frac{\Delta}{4}\right)$$

We first bound the mask term $\frac{1}{m}\|v_j\|_2^2$. Define the population average

$$d_j := \frac{1}{n}\|\mu_{j^\star} - \mu_j\|_2^2, \qquad \widehat{d}_j := \frac{1}{m}\|P_\Omega(\mu_{j^\star} - \mu_j)\|_2^2 = \frac{1}{m}\|v_j\|_2^2.$$

Each summand $(\mu_{j^\star,i} - \mu_{j,i})^2$ lies in $[0, 4]$. Since $\Omega$ is a uniform sample without replacement, Hoeffding's inequality (sampling without replace version, see Hoeffding (1963) or Proposition 1.2 of Bardenet & Maillard (2015)) yields, for any $t > 0$,

$$\mathbb{P}(|\widehat{d}_j - d_j| \geq t) \leq 2\exp\left(-\frac{mt^2}{8}\right).$$

Taking $t = \Delta/2$ and using $d_j \geq \Delta$ from (10) gives

$$\mathbb{P}(\widehat{d}_j \leq \Delta/2) \leq 2\exp\left(-\frac{m\Delta^2}{32}\right).$$

Now we bound the noise term. Conditional on $\Omega$, $\langle \eta, v_j \rangle$ is Gaussian with mean 0 and variance $(\sigma^2 + \tau^2)\|v_j\|_2^2$. Hence, for any $a > 0$, the classical Gaussian concentration bound implies

$$\mathbb{P}\left(\left|\frac{2}{m}\langle \eta, v_j \rangle\right| \geq a \,\Big|\, \Omega\right) \leq 2\exp\left(-\frac{m^2 a^2}{8(\sigma^2 + \tau^2)\|v_j\|_2^2}\right).$$

Using the coordinate bound $|(\mu_{j^\star,i} - \mu_{j,i})| \leq 2$ gives $\|v_j\|_2^2 \leq 4m$, so

$$\mathbb{P}\left(\left|\frac{2}{m}\langle \eta, v_j \rangle\right| \geq a \,\Big|\, \Omega\right) \leq 2\exp\left(-\frac{ma^2}{32(\sigma^2 + \tau^2)}\right).$$

Taking $a = \Delta/4$, on the event $\{\widehat{d}_j \geq \Delta/2\}$ and $\{|\frac{2}{m}\langle \eta, v_j \rangle| \leq \Delta/4\}$, the quantity

$$\frac{1}{m}\|v_j\|_2^2 + \frac{2}{m}\langle \eta, v_j \rangle$$

in (11) is at least $\Delta/4$, so

$$\mathbb{P}\left(\left(\frac{1}{m}\|v_j\|_2^2 + \frac{2}{m}\langle \eta, v_j \rangle\right) \geq \frac{\Delta}{4}\right) \geq 1 - 2\exp\left(-\frac{m\Delta^2}{32}\right) - 2\exp\left(-\frac{ma^2}{32(\sigma^2 + \tau^2)}\right).$$

A union bound over $j \neq j^\star$ completes the proof.

$\square$

# B. Details of ADAMSPHERE and HOLDOUTTOPK

**ADAMSPHERE optimizer.**    When optimizing the initial noise variable $z$ in diffusion-based inverse problems, we enforce the spherical constraint $\|z\|_2 = r$ (with $r$ fixed, or set to $\|z^{(0)}\|_2$ at initialization). This prevents the optimizer from trivially changing the noise magnitude and keeps the search within the typical set of the Gaussian prior. ADAMSPHERE optimizer adapts Adam to this constraint: at each iteration, we first project the Euclidean gradient onto the tangent space of the sphere at the current iterate, apply Adam's moment updates using this tangent gradient, and then project the preconditioned search direction back to the tangent space. Finally, we take a step along this tangent direction and retract back to the sphere.

We use the simple normalization retraction by default (rescale the updated vector to norm $r$), and also consider an exponential-map retraction that moves along a great circle on the sphere (on the sphere $\mathbb{S}_r^{d-1} = \{z : \|z\|_2 = r\}$, for a tangent direction $u$ satisfying $\langle u, z \rangle = 0$, the exponential-map retraction with step size $\eta$ is

$$\mathrm{Exp}_z(\eta u) = z\cos\left(\frac{\eta\|u\|_2}{r}\right) + r\frac{u}{\|u\|_2}\sin\left(\frac{\eta\|u\|_2}{r}\right),$$

with the case $u = 0$ defined by continuity as $\mathrm{Exp}_z(0) = z$.)

**HOLDOUTTOPK early stopping.**    When optimizing the initial noise, later iterates can begin to overfit the observed pixels: the measurement error keeps decreasing, but visual quality (or metrics such as PSNR/LPIPS) may deteriorate. To avoid overfitting, we use a simple holdout-based early-stopping rule, HOLDOUTTOPK. We randomly split the observed index set $\Omega$ into a *fit* subset $\Omega_{\mathrm{fit}}$ used for optimization and a disjoint *holdout* subset $\Omega_{\mathrm{ho}}$ that is never used in gradient updates. At

*Table 6.* Cross-Domain results for Gaussian deblurring and nonlinear deblurring measurements. The "Model" column indicates the source domain of the pretrained diffusion model. The "CelebA," "Bedroom," and "Church" columns indicate the target domains of the images being reconstructed. DPS always uses an in-domain model, i.e., the source and target domains coincide.

| Task | Method | Model | CelebA | | | Bedroom | | | Church | | |
|------|--------|-------|--------|--|--|---------|--|--|--------|--|--|
| | | | PSNR ↑ | LPIPS ↓ | SSIM ↑ | PSNR ↑ | LPIPS ↓ | SSIM ↑ | PSNR ↑ | LPIPS ↓ | SSIM ↑ |
| Gaussian | DPS | In-domain | 27.55 | **0.19** | 0.76 | 23.94 | 0.31 | 0.67 | 20.66 | **0.30** | 0.55 |
| | Ours | CelebA | **29.98** | 0.21 | **0.82** | 25.02 | 0.41 | 0.69 | 22.21 | 0.45 | 0.58 |
| | | Bedroom | 27.97 | 0.27 | 0.77 | **25.28** | **0.30** | **0.72** | 22.09 | 0.38 | 0.59 |
| | | Church | 27.66 | 0.29 | 0.75 | 24.76 | 0.35 | 0.69 | **22.43** | 0.30 | **0.62** |
| Nonlinear | DPS | In-domain | 24.88 | **0.26** | 0.71 | 22.51 | **0.38** | 0.63 | 20.31 | **0.39** | 0.54 |
| | Ours | CelebA | **25.21** | 0.27 | **0.74** | 22.34 | 0.47 | 0.63 | 20.52 | 0.51 | 0.52 |
| | | Bedroom | 24.59 | 0.39 | 0.69 | **22.67** | 0.39 | **0.64** | 20.79 | 0.47 | 0.54 |
| | | Church | 24.73 | 0.39 | 0.69 | 22.59 | 0.42 | 0.63 | **20.88** | 0.39 | **0.56** |

*Table 7.* Comparison with DMPlug under in-domain priors for Gaussian deblurring and super-resolution task.

| Task | Dataset | Ours | | | DMplug | | |
|------|---------|------|--|--|--------|--|--|
| | | PSNR ↑ | LPIPS ↓ | SSIM ↑ | PSNR ↑ | LPIPS ↓ | SSIM ↑ |
| Gaussian | CelebA | **29.98** | **0.21** | **0.82** | 29.52 | 0.23 | 0.81 |
| | Church | **22.43** | **0.30** | **0.62** | 22.24 | 0.32 | 0.61 |
| | Bedroom | **25.28** | **0.30** | **0.72** | 25.08 | 0.33 | 0.71 |
| Super-Res | CelebA | **31.271** | **0.184** | **0.863** | 31.122 | 0.198 | 0.857 |
| | Church | 23.088 | **0.266** | **0.674** | **23.183** | 0.280 | 0.667 |
| | Bedroom | **26.587** | **0.249** | **0.777** | 26.309 | 0.280 | 0.767 |

each iteration $t$, we evaluate a holdout score (e.g., the MSE on $\Omega_{\text{ho}}$) and keep the $K$ iterates with the lowest holdout scores. We then output an iterate chosen from this top-$K$ set (e.g., the best within the top-$K$, or uniformly at random among them).

Our idea is inspired by standard practice in statistical machine learning, where a validation set is used to approximate test performance and guide model selection. However, unlike standard model selection that picks the single lowest-error candidate, we keep the top-$K$ lowest-error iterates and eventually return the latest iterate among these top $K$. Using $K > 1$ stabilizes selection by avoiding the risk of picking a single spurious "best" iterate due to holdout noise, while still favoring iterates that consistently generalize well to unseen pixels. In practice, HOLDOUTTOPK provides a lightweight and robust way to reduce overfitting.

## C. Additional experiment results and experiment details

### C.1. Cross-Domain Inverse Problem Solving

Table 6 presents additional cross-domain results for Gaussian and nonlinear deblurring. Consistent with Section 4.1, these experiments further confirm that a weak diffusion prior, such as a few-step or domain-mismatched one, remains highly competitive with the DPS baseline. For Gaussian deblurring, DPS performs well in-domain, but our method frequently surpasses it in PSNR and SSIM despite using substantially weaker priors. With the more challenging nonlinear deblurring task, the same qualitative conclusions hold and further demonstrate that weak priors are effective and robust for inverse problems.

### C.2. Comparison with Optimization-Based Methods

Table 7 reports further comparisons between our method and DMPlug on Gaussian deblurring and $4\times$ super-resolution. Across both tasks and all three datasets, our method matches or outperforms DMPlug in most metrics. For Gaussian deblurring, our method improves PSNR on all three datasets, with consistently competitive LPIPS and SSIM. For super-resolution, the two methods are close, while our method generally yields lower LPIPS and higher SSIM.

| Task | Dataset | DAPS | | | DiffPIR | |
|------|---------|------|------|------|------|------|
| | | PSNR ↑ | SSIM ↑ | LPIPS ↓ | PSNR ↑ | LPIPS ↓ |
| Inpainting | CelebA-HQ | 32.35 | 0.869 | 0.112 | 31.96 | 0.192 |
| Inpainting | Bedroom | 28.93 | 0.845 | 0.142 | 28.12 | 0.247 |
| Inpainting | Church | 24.95 | 0.781 | 0.145 | 24.81 | 0.218 |
| Gaussian deblur | CelebA-HQ | 30.08 | 0.794 | 0.137 | 27.35 | 0.206 |
| Gaussian deblur | Bedroom | 26.64 | 0.733 | 0.239 | 23.84 | 0.341 |
| Gaussian deblur | Church | 23.54 | 0.646 | 0.256 | 21.01 | 0.317 |
| SR 4× | CelebA-HQ | 30.21 | 0.802 | 0.142 | 29.59 | 0.177 |
| SR 4× | Bedroom | 26.68 | 0.740 | 0.236 | 25.98 | 0.280 |
| SR 4× | Church | 23.57 | 0.656 | 0.248 | 22.66 | 0.277 |

*Table 8.* Comparison against DAPS and DiffPIR.

### C.3. Comparison with additional diffusion-solver baseline: DAPS and DiffPIR

We include DAPS and DiffPIR as additional diffusion-based inverse-problem baselines. This comparison is intended to check whether the conclusions in the main text are specific to using DPS as the full-prior baseline. We consider three inverse tasks: random inpainting, Gaussian deblurring, and $4\times$ super-resolution, evaluated on CelebA-HQ, LSUN Bedroom, and LSUN Church. Table 8 reports the results.

DAPS is a strong baseline across these tasks, especially on perceptual metrics, while DiffPIR is generally weaker under this setup. These results provide additional context for the baseline landscape. They do not change the main message of the paper. Our early-stopped initial-noise optimization (INO) under weak priors achieves competitive PSNR across tasks. For example, on CelebA-HQ inpainting, INO reaches a PSNR of 33.78 (see Table 3), outperforming DAPS (32.35) and DiffPIR (31.96). The out-of-domain setting also remains competitive: using a Church-trained prior to recover CelebA-HQ images still achieves a PSNR of 32.62.

### C.4. Random-network weak prior: Deep Random Projector

We include Deep Random Projector (DRP) (Li et al., 2023) as an additional baseline for random inpainting. DRP is closely related to our setting because it also solves inverse problems through an optimization-based generative prior. However, it represents an even weaker prior than the weak diffusion priors studied in the main text: instead of using a pretrained diffusion generator, DRP uses a randomly initialized frozen network. Thus, it does not use any learned data distribution and relies only on the architectural inductive bias of the generator.

We evaluate DRP on random inpainting for CelebA-HQ, LSUN Bedroom, and LSUN Church, using the same setting as Appendix C.12: 70% random masking with Gaussian noise $\sigma = 0.01$ on the observed pixels. All results are averaged over 100 test images per dataset. Table 9 reports the results.

DRP obtains reasonable reconstructions, showing that architectural inductive bias alone can provide useful regularization for inverse problems.

However, it is consistently worse than our weak diffusion prior. In particular, DRP reconstructions are noticeably blurrier and contain weaker fine-scale details; see Figure 13 for an example. Compared with the in-domain 3-step diffusion prior in Table 3, DRP is lower by 4.69 dB on CelebA-HQ, 3.47 dB on Bedroom, and 2.14 dB on Church. Its SSIM is also lower, especially on CelebA-HQ and Bedroom. These results support the view that weak diffusion priors are not arbitrary weak generators: even when their unconditional samples are low-fidelity, they still carry useful information from pretraining that is absent from a randomly initialized network.

### C.5. Scientific inverse problems

We further evaluate prior mismatch on scientific inverse problems from InverseBench (Zheng et al., 2025). These experiments are useful because, in scientific imaging, a well-matched pretrained generative prior may be unavailable, so one may need to reuse a prior trained for a different physical system.

| Dataset | PSNR (dB) ↑ | PSNR std | SSIM ↑ | SSIM std |
|---|---|---|---|---|
| CelebA-HQ | 29.09 | 1.64 | 0.8599 | 0.0262 |
| Bedroom | 25.41 | 2.44 | 0.7831 | 0.0474 |
| Church | 22.79 | 2.30 | 0.7425 | 0.0571 |

*Table 9.* Deep Random Projector (DRP) performance on random inpainting with 70% random masking and Gaussian noise $\sigma = 0.01$ on observed pixels. Results are averaged over 100 images per dataset using a randomly initialized frozen SkipNet.

| Number of receivers | INO | DPS | DAPS |
|---|---|---|---|
| 60 | 21.97 | 20.29 | 21.33 |
| 180 | 27.66 | 26.24 | 25.53 |
| 360 | 27.99 | 26.46 | 26.68 |

*Table 10.* PSNR comparison on the Linear Scattering inverse problem under prior mismatch. We use the FWI prior from InverseBench as a mismatched prior and evaluate three receiver settings. Higher PSNR is better.

**Linear scattering with an FWI prior.** We first consider the Linear Scattering problem. Following the InverseBench setup, we evaluate three measurement settings with 60, 180, and 360 receivers, corresponding to different levels of measurement informativeness. We use the Full Waveform Inversion (FWI) prior provided by InverseBench as a mismatched prior. This prior is strongly mismatched to the Linear Scattering task, since it is trained for recovering subsurface physical properties from full waveform measurements. We compare DPS, DAPS, and our initial-noise optimization (INO), with all methods lightly tuned.

Table 10 reports PSNR. Compared with the matched-prior results in Table 3 of Zheng et al. (2025), the matched-prior setting remains stronger overall. Nevertheless, the degradation under prior mismatch is moderate rather than catastrophic. With 180 and 360 receivers, the reconstructions remain reasonable; with 60 receivers, INO even attains higher PSNR than several strong-prior methods reported in InverseBench. Among the mismatched-prior methods tested here, INO consistently outperforms both DPS and DAPS across all three receiver settings. The gains over DPS are 1.68, 1.42, and 1.53 dB for 60, 180, and 360 receivers, respectively; the gains over DAPS are 0.64, 2.13, and 1.31 dB.

**Nonlinear black hole imaging.** We also evaluate the nonlinear Black Hole Imaging problem from InverseBench. We compare a matched prior with a mismatched prior, where the mismatched prior is taken from the Linear Scattering model and resized to the Black Hole Imaging resolution. Table 11 reports PSNR and blur PSNR.

Under the matched prior, DPS gives the best PSNR, followed by INO and DAPS. Under prior mismatch, however, INO becomes the strongest method among the three: it improves over DPS and DAPS by 2.04 dB and 2.82 dB in PSNR, respectively, and by 2.41 dB and 3.48 dB in blur PSNR. INO also has the smallest degradation from the matched-prior setting to the mismatched-prior setting.

At the same time, these results should be interpreted with some care. INO is computationally slower than DPS and DAPS because it solves an iterative optimization problem over the initial noise. In addition, although INO performs well in PSNR and blur PSNR, other physics-aware metrics such as $\chi^2$ are less favorable in our experiments. This suggests that, for scientific inverse problems, optimizing the initial noise with a simple pixel- or image-space reconstruction loss may not be enough. Better task-specific loss functions, including losses tied to the forward model or physics-based statistics such as $\chi^2$, may be needed to align INO with the evaluation criteria used in scientific imaging.

These scientific inverse-problem experiments support the main message of the paper: weak priors can still give useful reconstructions when the measurements are informative. At the same time, the gap between matched and mismatched priors appears larger here than in our natural-image experiments. One possible reason is that different scientific domains may share less local spatial structure than natural-image domains, so a mismatched prior has less transferable structural information to propagate from the measurements. These results also point to an important direction for future work: for scientific inverse problems, the optimization objective for initial-noise optimization should be designed to match the scientific task and evaluation metric, rather than relying only on generic image-space reconstruction losses.

| Method | PSNR (Mismatch) | PSNR (Matched) | Blur PSNR (Mismatch) | Blur PSNR (Matched) |
|--------|-----------------|----------------|----------------------|---------------------|
| DPS | $18.06 \pm 3.00$ | $26.15 \pm 4.82$ | $21.98 \pm 4.18$ | $33.15 \pm 6.87$ |
| DAPS | $17.28 \pm 2.14$ | $22.56 \pm 3.81$ | $20.91 \pm 2.80$ | $26.89 \pm 4.55$ |
| INO | $20.10 \pm 2.93$ | $24.15 \pm 3.15$ | $24.39 \pm 3.87$ | $29.32 \pm 4.49$ |

*Table 11.* Results on nonlinear Black Hole Imaging under matched and mismatched priors. The mismatched prior is taken from the Linear Scattering model and resized to the Black Hole Imaging resolution. Higher PSNR and blur PSNR are better.

*Table 12.* Failure mode results for box inpainting.

| Mask | Method | PSNR ↑ | LPIPS ↓ | SSIM ↑ |
|------|--------|--------|---------|--------|
| | DPS | 25.00 | **0.22** | 0.79 |
| $0.3 \times 0.3$ | Ours (In) | **25.28** | 0.23 | **0.81** |
| | Ours (Out) | 22.42 | 0.32 | 0.77 |
| | DPS | 22.28 | **0.24** | 0.76 |
| $0.4 \times 0.4$ | Ours (In) | **22.31** | 0.25 | **0.78** |
| | Ours (Out) | 19.20 | 0.35 | 0.73 |
| | DPS | **19.38** | **0.27** | 0.71 |
| $0.5 \times 0.5$ | Ours (In) | 19.09 | 0.30 | **0.72** |
| | Ours (Out) | 16.81 | 0.39 | 0.67 |
| | DPS | **17.72** | **0.30** | 0.65 |
| $0.6 \times 0.6$ | Ours (In) | 17.27 | 0.34 | **0.66** |
| | Ours (Out) | 15.35 | 0.43 | 0.61 |

### C.6. Failure Modes

Table 12 provides the full quantitative results for box inpainting, where the goal is to reconstruct Church images while masking increasingly large square regions. For moderate mask sizes ($0.3 \times 0.3$ and $0.4 \times 0.4$), our in-domain weak prior remains competitive with DPS. However, as the masked region grows to $0.5 \times 0.5$ and $0.6 \times 0.6$, DPS begins to dominate in both PSNR and LPIPS, while our in-domain variant exhibits noticeable degradation, as illustrated in the additional visualizations in Figure 14. This confirms the conclusion that weak diffusion priors remain effective for reconstruction-focused inverse problems but encounter fundamental limitations in generation-heavy settings.

We observe a similar trend for large-scale super-resolution. Qualitative results for $16\times$ super-resolution are shown in Figure 15, where out-of-domain reconstructions generated by a CelebA-pretrained model begin to exhibit face-like structures when applied to Church images.

### C.7. Latent Diffusion Applications

We evaluate two few-step priors based on Stable Diffusion 2.1 and DiT, and compare against two baselines: DPS (Chung et al., 2023) and DSG (Yang et al., 2024), all implemented with SD 2.1 backbones.

Table 13 summarizes ImageNet results for inpainting and Gaussian deblurring using latent diffusion models. Overall, our method performs comparably to DPS on both tasks and substantially outperforms DSG. Even with much weaker generative models, our framework preserves strong reconstruction quality for both inpainting and deblurring. Visual comparisons are provided in Figure 12 (Gaussian deblurring) and Figure 11 (inpainting).

### C.8. Spatial autocorrelation diagnostic

We provide the details of the spatial autocorrelation diagnostic used in Section 3.3. The goal is to measure whether weak and stronger diffusion priors preserve similar local pixel-level structure, even when their unconditional samples differ in fidelity or semantic content.

Let $x \in \mathbb{R}^{H \times W \times C}$ be a generated image, where $C = 3$ for RGB images. For each channel $c$, let $x_{c,p}$ denote the pixel value

*Table 13.* Results for inpainting and Gaussian deblurring. We report ES metrics when available; otherwise, final results are used. Best results are in **bold**, second-best are underlined.

| Task | Method | Model | PSNR ↑ | LPIPS ↓ | SSIM ↑ |
|------|--------|-------|--------|---------|--------|
| Inpainting | Ours | DiT | **28.44** | 0.394 | 0.727 |
| | | SD2.1 | 26.82 | 0.374 | 0.745 |
| | DPS | | 28.21 | **0.292** | **0.797** |
| | DSG | SD2.1 | 23.33 | 0.503 | 0.597 |
| Gaussian | Ours | DiT | **26.48** | 0.458 | 0.678 |
| | | SD2.1 | 25.40 | 0.443 | 0.685 |
| | DPS | | 25.49 | **0.388** | **0.728** |
| | DSG | SD2.1 | 22.09 | 0.510 | 0.539 |

| | CelebA 3-step | CelebA 20-step | Bedroom 3-step | Bedroom 20-step |
|------|------|------|------|------|
| CelebA 3-step | 1.0000 | 0.9998 | 0.9989 | 0.9965 |
| CelebA 20-step | 0.9998 | 1.0000 | 0.9987 | 0.9967 |
| Bedroom 3-step | 0.9989 | 0.9987 | 1.0000 | 0.9993 |
| Bedroom 20-step | 0.9965 | 0.9967 | 0.9993 | 1.0000 |

*Table 14.* Pairwise Pearson correlations between spatial autocorrelation profiles over representative lags $\{1, 2, 4, 8, 16, 32\}$.

at spatial location $p \in \Omega = \{1, \ldots, H\} \times \{1, \ldots, W\}$, and let

$$\bar{x}_c = \frac{1}{|\Omega|} \sum_{p \in \Omega} x_{c,p}.$$

For an integer lag $r \geq 0$, define

$$\mathcal{P}_r = \big\{(p,q) \in \Omega \times \Omega : \mathrm{round}(\|p - q\|_2) = r\big\}.$$

The normalized spatial autocorrelation of image $x$ at lag $r$ is

$$C_x(r) = \frac{1}{C} \sum_{c=1}^{C} \frac{|\mathcal{P}_r|^{-1} \sum_{(p,q) \in \mathcal{P}_r} (x_{c,p} - \bar{x}_c)(x_{c,q} - \bar{x}_c)}{|\Omega|^{-1} \sum_{p \in \Omega} (x_{c,p} - \bar{x}_c)^2}.$$

By construction, $C_x(0) = 1$. For each prior and sampler setting, we generate 100 unconditional samples and report the averaged profile

$$C(r) = \frac{1}{100} \sum_{i=1}^{100} C_{x_i}(r).$$

We compare DDPM priors trained on CelebA-HQ and LSUN Bedroom, sampled with either 3 or 20 DDIM steps at resolution $256 \times 256$. For each setting, we average over samples, RGB channels, and all pixel-pair directions with the same rounded Euclidean distance. Table 2 reports representative lags. Pairwise correlation matrix is reported in Table 14.

The profiles are highly similar across both axes of variation: sampler strength and training domain. For example, the Pearson correlation between the 3-step Bedroom profile and the 20-step CelebA-HQ profile is $0.9987$. This supports the interpretation that weak natural-image priors may lose sample fidelity or domain match while still preserving short-range local structural information.

## C.9. DPS with mismatched priors

To separate the effect of domain mismatch from the choice of inverse solver, we also evaluate DPS with matched and mismatched full diffusion priors. In the main experiments, DPS uses a full diffusion prior trained on the target domain. Here, we keep the solver fixed as DPS, but vary the source domain of the pretrained diffusion model. This isolates the domain-match axis from the sampler-strength axis.

| Source model | Target dataset | PSNR ↑ | LPIPS ↓ | SSIM ↑ |
|---|---|---|---|---|
| CelebA-HQ | CelebA-HQ | 24.88 | 0.264 | 0.709 |
| Church | CelebA-HQ | 22.38 | 0.433 | 0.631 |
| Bedroom | CelebA-HQ | 23.57 | 0.411 | 0.660 |
| Church | Church | 20.31 | 0.385 | 0.539 |
| CelebA-HQ | Church | 20.06 | 0.508 | 0.490 |
| Bedroom | Church | 19.84 | 0.469 | 0.512 |
| Bedroom | Bedroom | 22.51 | 0.375 | 0.633 |
| CelebA-HQ | Bedroom | 21.98 | 0.477 | 0.588 |
| Church | Bedroom | 22.04 | 0.422 | 0.614 |

*Table 15.* Performance of DPS with matched and mismatched full diffusion priors on nonlinear deblurring. The source model indicates the domain of the pretrained diffusion prior, and the target dataset indicates the reconstruction domain. Matched-prior rows are included as references.

Table 15 reports results for the nonlinear deblurring task. The matched-prior rows are included as references. As expected, the matched prior usually gives the best performance for each target dataset. However, the degradation under prior mismatch is not catastrophic in all cases. For Church and Bedroom targets, mismatched priors remain close to the matched DPS baseline in PSNR, while CelebA-HQ is more sensitive to source-domain mismatch. These results support the view that mismatched priors can still provide useful reconstruction performance.

### C.10. Sensitivity to measurement noise

We further study how measurement noise affects prior sensitivity. We vary the Gaussian noise level $\sigma \in \{0.01, 0.05, 0.2, 0.5, 1.0\}$ and compare DPS, which uses a strong in-domain prior, with our initial-noise optimization (INO) method using weak priors in both in-domain and out-of-domain settings. Table 16 reports PSNR, LPIPS, SSIM, and the selected early-stopping epoch for INO.

The results show that INO is highly competitive at low and moderate noise levels. For $\sigma \leq 0.5$, in-domain INO achieves the best PSNR and SSIM, and the out-of-domain INO variant also remains useful. However, as the noise level increases, the gap between in-domain and out-of-domain INO becomes larger. At $\sigma = 1.0$, DPS with a strong in-domain prior outperforms both INO variants. This is consistent with our failure-mode picture: when measurements are noisy, they provide fewer reliable local anchors, and reconstruction becomes more dependent on the strength and domain match of the prior.

### C.11. On computational comparability with baseline algorithms

INO and DMPlug both use the same initial-noise optimization framework, so we use the exact same optimization budget in all comparisons to ensure fairness.

DPS and INO allocate computation in fundamentally different ways, so a strict compute-matched comparison is not straightforward. More precisely

- DPS has an essentially fixed per-run cost once the number of reverse diffusion steps is set: each reverse step involves roughly one denoiser evaluation, so the total cost is driven by the chain length. Moreover, existing studies (Ma et al., 2025) suggest that increasing the chain length scales poorly, so the DPS budget is typically treated as effectively fixed. People typically choose 500 or 1000 reverse steps depending on the task.

- INO has a tunable compute budget through the number of initial-noise optimization iterations. Each iteration requires running a $K$-step DDIM generation, so the dominant cost scales with the product of the iteration count and the DDIM step count. In our experiments, we typically use a 3-step DDIM and 500 iterations of optimization.

In our experiments in Section 4, a single run of INO costs about $1.5\times$ to $3\times$ as much compute as a single DPS run. This comparison is still suitable for our main purpose, which is to study the effect of the prior. Specifically, our goal is to test whether weak priors can solve inverse problems in data-informative regimes, rather than to claim that INO is uniformly more efficient or uniformly better than DPS. We therefore focus on qualitative behavior and reconstruction metrics such as PSNR and SSIM, instead of treating the main comparison as a strictly compute-matched efficiency benchmark. Moreover,

| Noise $\sigma$ | Method | Domain | PSNR ↑ | LPIPS ↓ | SSIM ↑ | Epoch |
|---|---|---|---|---|---|---|
| 0.01 | DPS | In-domain | 31.98 | **0.1447** | 0.8827 | – |
| 0.01 | INO | In-domain | **33.78** | 0.1476 | **0.9147** | 974.09 |
| 0.01 | INO | Out-of-domain | 32.62 | 0.1626 | 0.8985 | 973.42 |
| 0.05 | DPS | In-domain | 30.12 | 0.1788 | 0.8403 | – |
| 0.05 | INO | In-domain | **33.39** | **0.1457** | **0.9096** | 940.61 |
| 0.05 | INO | Out-of-domain | 32.08 | 0.1676 | 0.8889 | 921.33 |
| 0.20 | DPS | In-domain | 26.93 | 0.2281 | 0.7602 | – |
| 0.20 | INO | In-domain | **30.41** | **0.1981** | **0.8478** | 358.95 |
| 0.20 | INO | Out-of-domain | 28.38 | 0.2862 | 0.7877 | 351.41 |
| 0.50 | DPS | In-domain | 24.45 | 0.2692 | 0.7004 | – |
| 0.50 | INO | In-domain | **26.30** | **0.2596** | **0.7335** | 168.64 |
| 0.50 | INO | Out-of-domain | 23.86 | 0.4366 | 0.6183 | 155.89 |
| 1.00 | DPS | In-domain | **22.42** | **0.3095** | **0.6518** | – |
| 1.00 | INO | In-domain | 20.52 | 0.3771 | 0.4915 | 119.05 |
| 1.00 | INO | Out-of-domain | 16.88 | 0.6171 | 0.2962 | 108.28 |

*Table 16.* Quantitative comparison under varying measurement noise levels $\sigma \in \{0.01, 0.05, 0.2, 0.5, 1.0\}$. We compare DPS with a strong in-domain prior against INO with weak priors in both in-domain and out-of-domain settings. Metrics follow standard conventions: PSNR/SSIM higher is better, and LPIPS lower is better. Epoch reports the selected early-stopping iteration for INO.

once the diffusion chain length is fixed, DPS does not have a directly comparable knob for using additional compute. For completeness, we also include an explicitly compute-matched comparison below.

Table 17 reports matched-cost comparisons on CelebA-HQ inpainting using both number of function evaluations (NFE) and wall-clock time. INO uses more memory than DPS, but it can use additional compute effectively. At about 1000 NFE, INO with 333 optimization iterations reaches 31.86 PSNR, close to DPS with 1000 NFE at 32.27 PSNR. At about 3000 NFE, INO reaches 34.13 PSNR, compared with 32.36 PSNR for DPS using the best of three runs. Even compared with DPS using the best of five runs, which uses 5000 NFE and more wall-clock time than INO, INO still achieves higher reconstruction quality. These results show that INO is more expensive than a single DPS run, but provides a controllable cost–quality tradeoff.

| Method | NFE | Runtime (s) | Peak alloc. (MB) | Peak reserved (MB) | PSNR ↑ | SSIM ↑ | LPIPS ↓ |
|---|---|---|---|---|---|---|---|
| INO (333 iters) | 999 | 56.34 | 7362.67 | 7826.00 | 31.86 | 0.8721 | 0.2039 |
| DPS (single run) | 1000 | 40.44 | 2629.25 | 2918.00 | 32.27 | 0.8851 | 0.1449 |
| INO (1000 iters) | 3000 | 171.16 | 7852.42 | 8474.00 | 34.13 | 0.9174 | 0.1485 |
| DPS (best of 3) | 3000 | 120.40 | 2629.25 | 2918.00 | 32.36 | 0.8855 | 0.1442 |
| DPS (best of 5) | 5000 | 200.04 | 2629.25 | 2918.00 | 32.40 | 0.8864 | 0.1436 |

*Table 17.* Matched-compute comparison on CelebA-HQ inpainting. Compute is measured by the number of function evaluations (NFE), and we also report wall-clock runtime and GPU memory usage. INO has higher memory cost than DPS, but it can use additional optimization iterations to improve reconstruction quality.

### C.12. Experiment Configuration

**Forward Operator** The forward operators tested are inpainting, Gaussian deblurring, super-resolution, and nonlinear deblurring, and the detailed setup is summarized in Table 18. All measurements have additive Gaussian noise with $\sigma = 0.01$. The nonlinear blur model follows Tran et al. (2021) available on public GitHub repository `https://github.com/VinAIResearch/blur-kernel-space-exploring`.

**Non-latent Diffusion** For non-latent diffusion settings, DPS was executed with 1000 sampling steps, while both DM-Plug and our proposed method employed 1000 optimization steps. All experiments used Hugging Face pretrained diffusion models on the Church, Bedroom, and CelebA-HQ datasets available with ID `google/ddpm-church-256`,

*Table 18.* Measurement operators and degradation models used in the experiments.

| Task | Operator | Parameters |
|---|---|---|
| Inpainting | Random masking | 70% of pixels removed |
| Gaussian Deblurring | Gaussian convolution | Kernel size $= 61$, intensity $= 3$ |
| Super Resolution | Downsampling | Factor $4\times$ |
| Nonlinear Deblurring | BKSE (GOPRO_wVAE.pth) | Pretrained GoPro wave model |

*Table 19.* Algorithm hyperparameters for each measurement task.

| Task | Learning Rate | $k$ for HOLDOUTTOPK |
|---|---|---|
| Inpainting | 0.02 | 90% / 10% |
| Gaussian Deblurring | 0.02 | 80% / 20% |
| Super Resolution | 0.01 | 95% / 5% |
| Nonlinear Deblurring | 0.01 | 90% / 10% |

`google/ddpm-bedroom-256`, and `google/ddpm-celebahq-256`. We adopted task-specific learning rates and HOLDOUTTOPK ratios as reported in Table 19. DPS and DMPlug are implemented under their default configs.

**Latent Diffusion**    For latent diffusion experiments, all baseline methods (DPS and DSG) used 500 steps. Our approach used 500 optimization steps. The learning rate for our approach was set to 0.1 for DiT and 0.2 for Stable Diffusion 2.1. All models were used with a null prompt as an empty text prompt or class 1000.

Baselines' implementation is available on the public GitHub repository `https://github.com/tongdaxu/diffusers-Diffusion-Posterior-Sampling` , which provides diffuser-based implementations of three algorithms with default configurations.

Stable Diffusion 2.1 is available on Hugging Face with ID *Manojb/stable-diffusion-2-1-base*. The DiT model is in the official release at `https://github.com/facebookresearch/DiT`, and we use the $(512 \times 512)$ resolution model.

All experiments were conducted on eight NVIDIA L40S GPUs. The most computationally intensive configuration, Stable Diffusion with optimization-based reconstruction, required approximately seven minutes to reconstruct a single image using 500 steps.

# D. Visualization

## D.1. Reconstruction Process with Different Priors

Figures 5–10 show results of our initial-noise optimization algorithm for recovering images using both in-distribution and out-of-distribution priors.

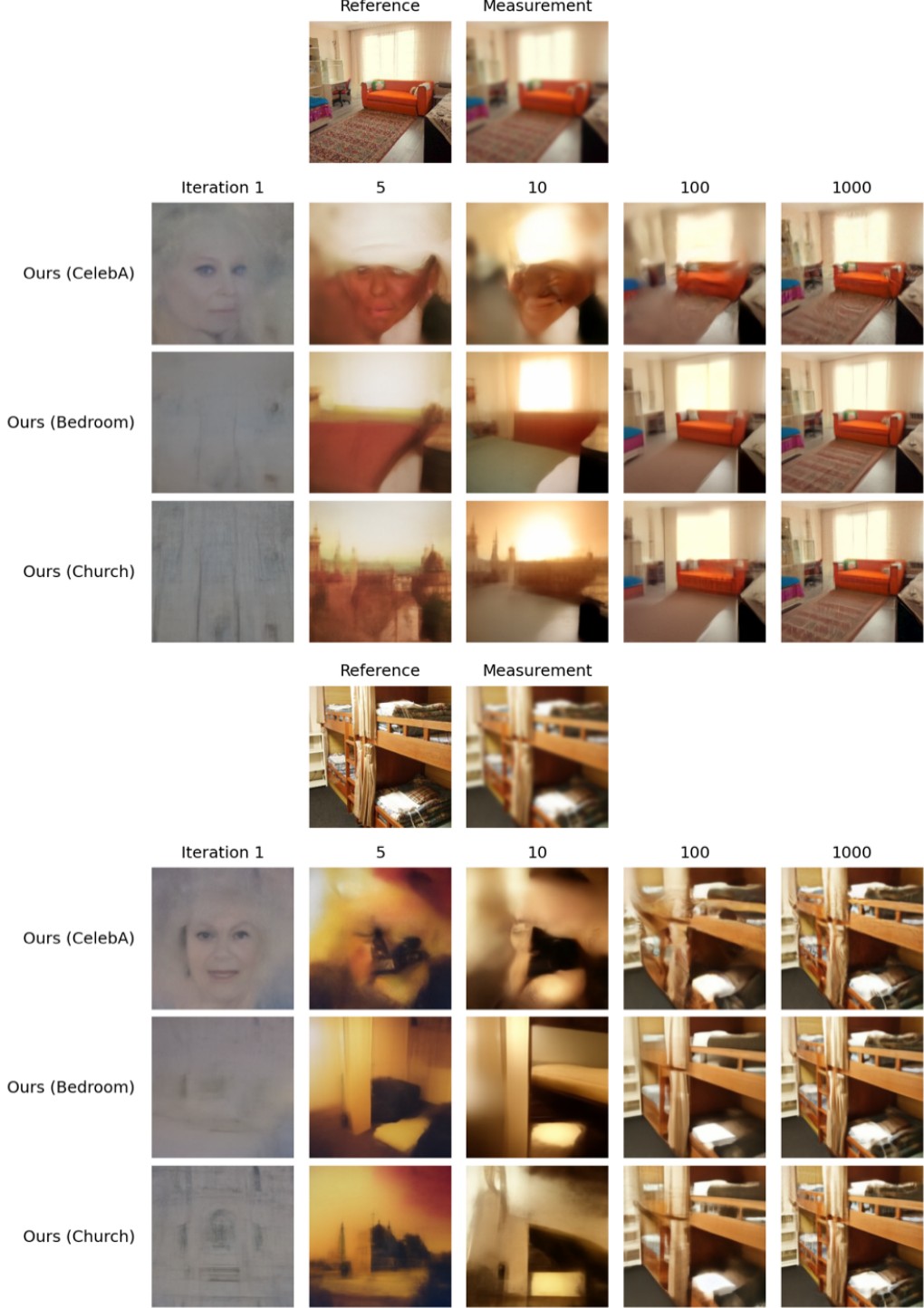

*Figure 5.* Bedroom reconstruction results for Gaussian deblurring.

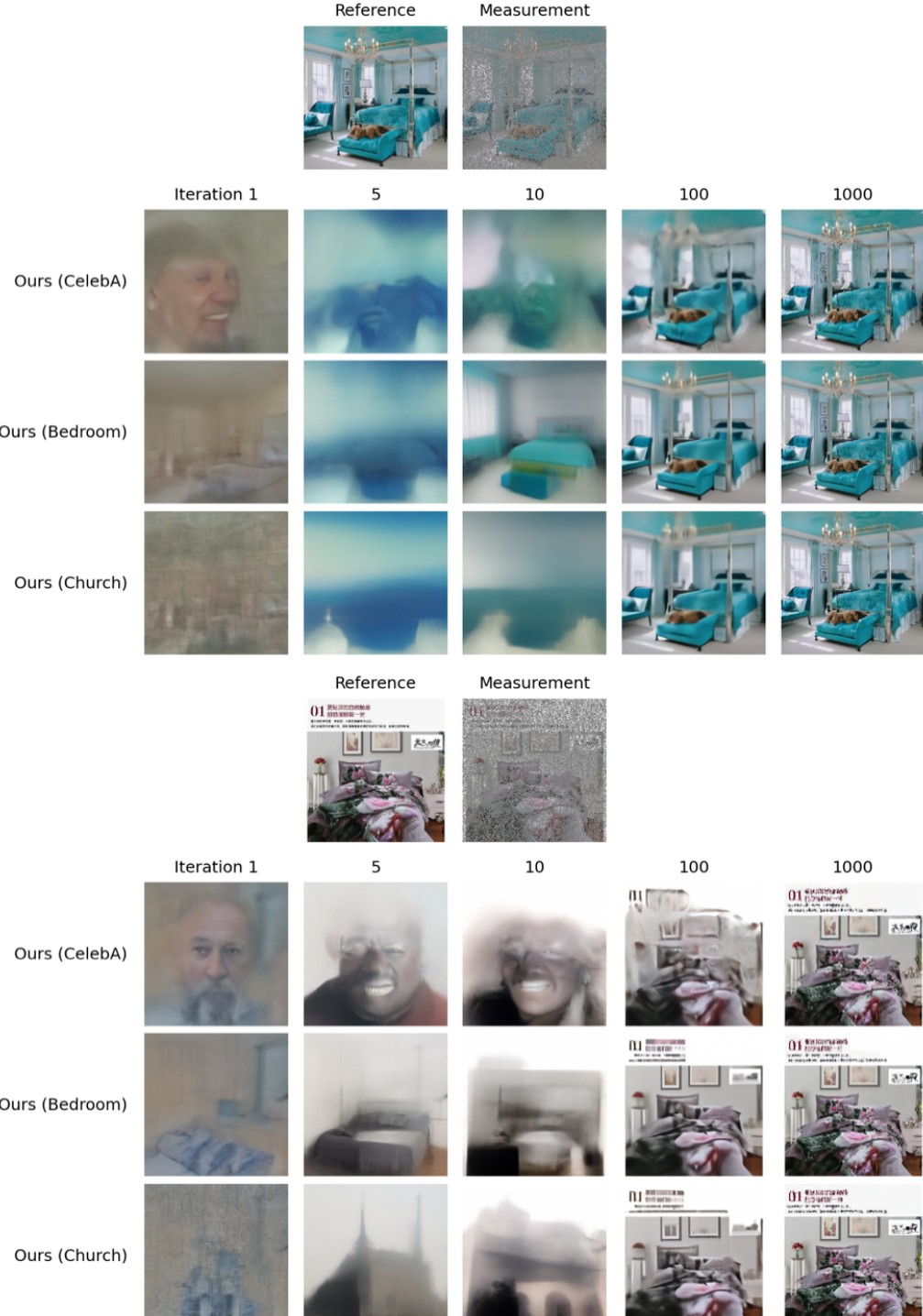

*Figure 6.* Bedroom reconstruction results for inpainting.

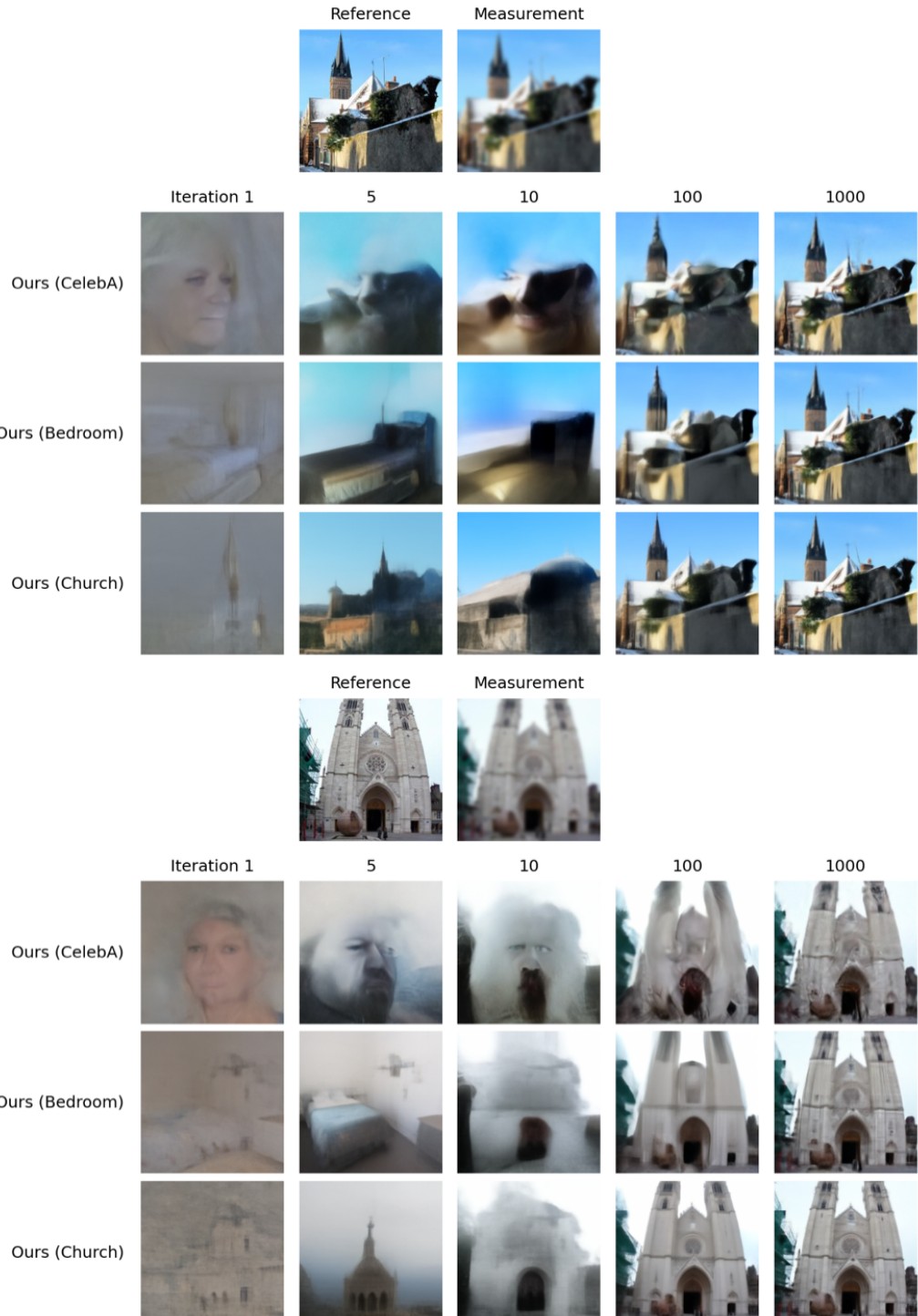

*Figure 7.* Church reconstruction results for Gaussian deblurring.

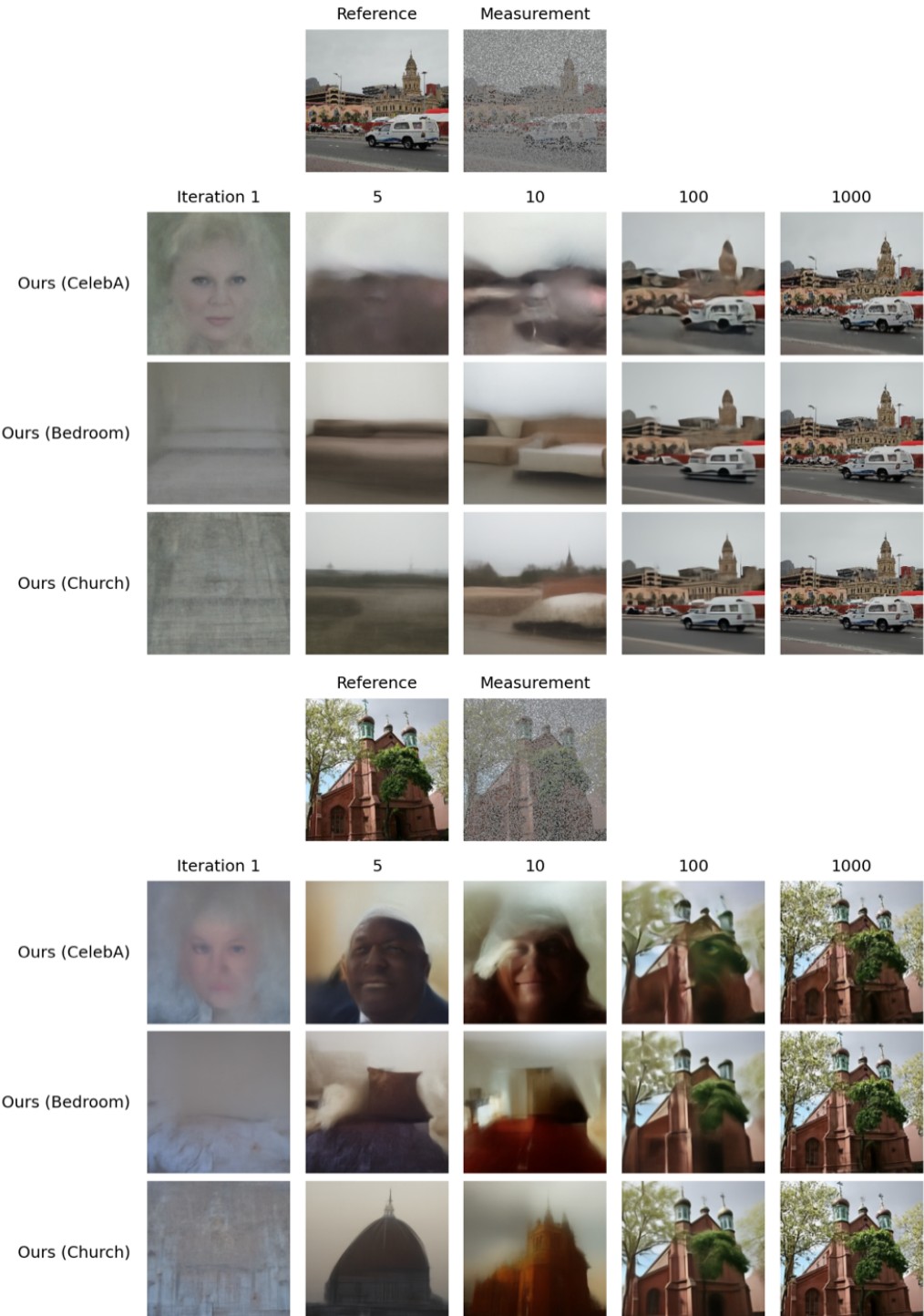

*Figure 8.* Church reconstruction results for inpainting.

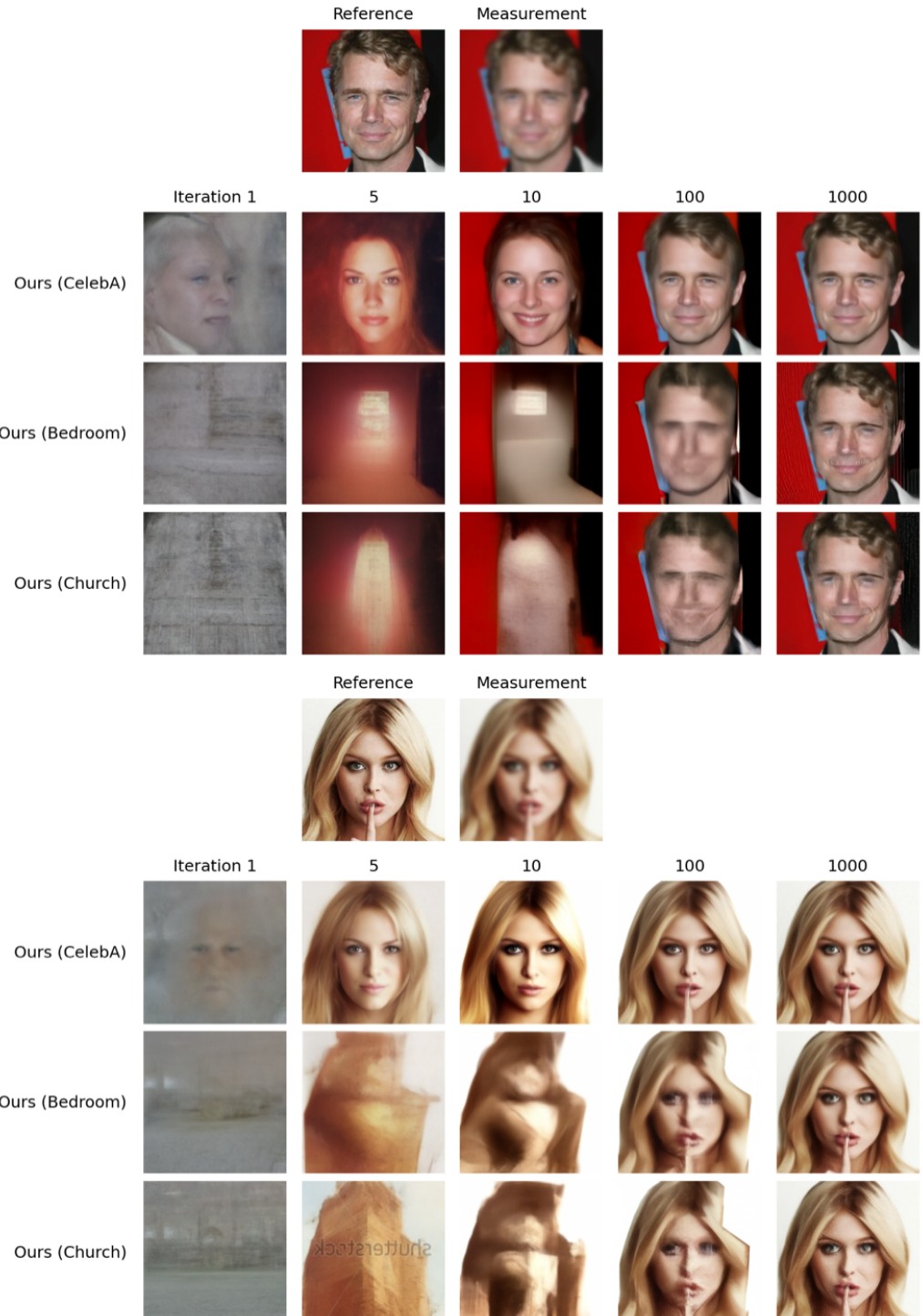

*Figure 9.* CelebA reconstruction results for Gaussian deblurring.

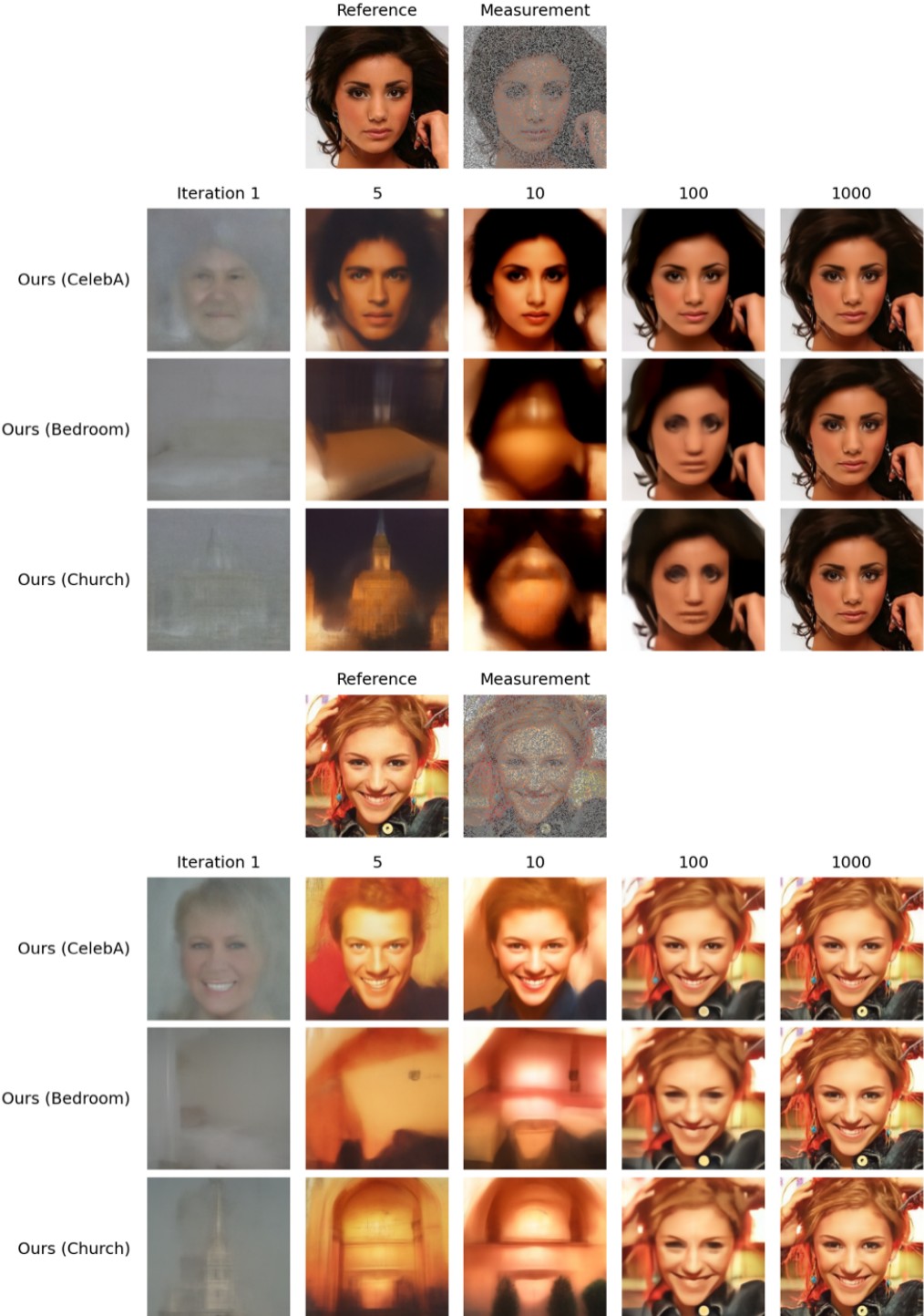

*Figure 10.* CelebA reconstruction results for inpainting.

## D.2. Reconstruction Result for Latent Diffusion Application

Figures 11 and 12 show reconstruction results for several latent diffusion algorithms. We implement our methods using both SD2.1 and DiT priors.

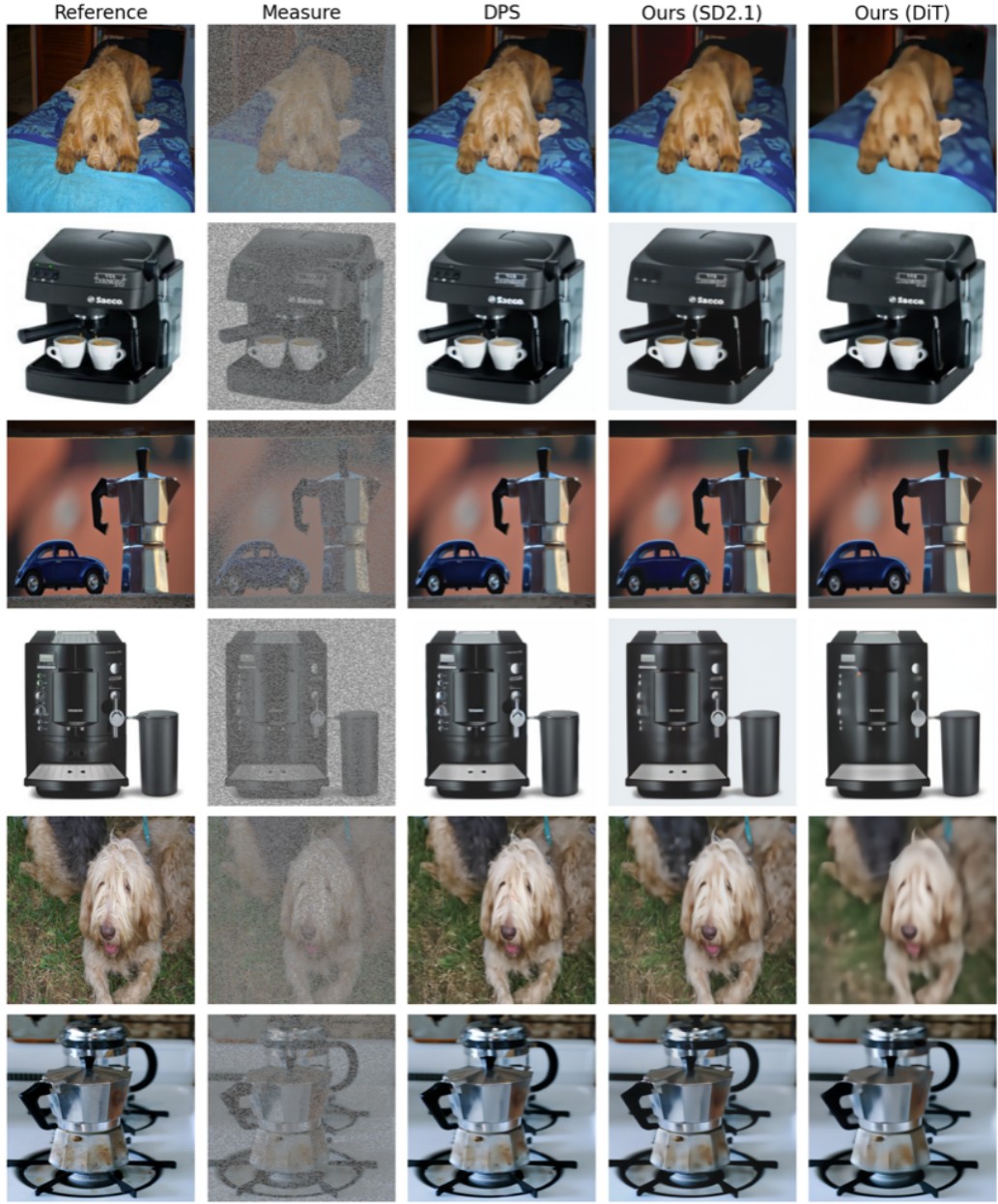

*Figure 11.* Reconstruction results for inpainting task on ImageNet using latent diffusion priors (Stable Diffusion 2.1 and DiT), compared with DPS

### D.3. Reconstruction using DRP prior

Figure 13 visualizes reconstructions obtained with the DRP prior. Because DRP uses a randomly initialized generator, its reconstructions are noticeably blurrier and contain weaker fine-scale details.

### D.4. Reconstruction Failure Case

Figures 14 and 15 visualize box inpainting and super-resolution results for DPS and our method. Our method is shown with both in-domain and out-of-domain weak priors. The failure mode is evident from the reconstruction inconsistencies.

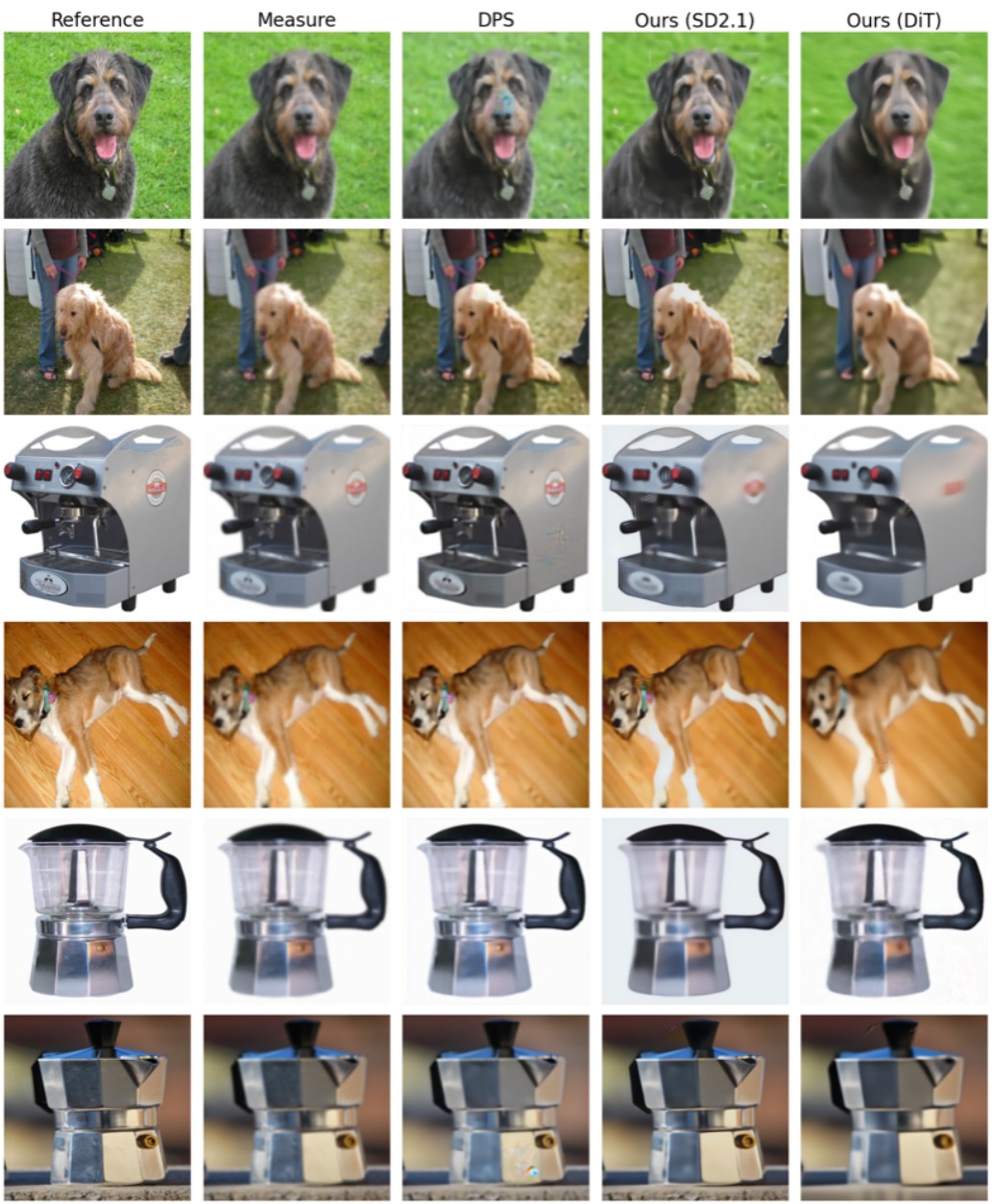

*Figure 12.* Reconstruction results for Gaussian deblurring task on ImageNet using latent diffusion priors (Stable Diffusion 2.1 and DiT), compared with DPS

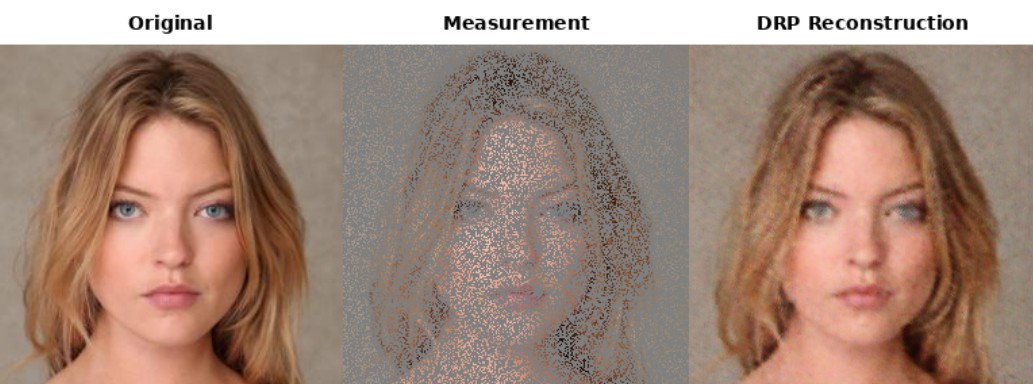

*Figure 13.* DRP reconstruction result for random inpainting.

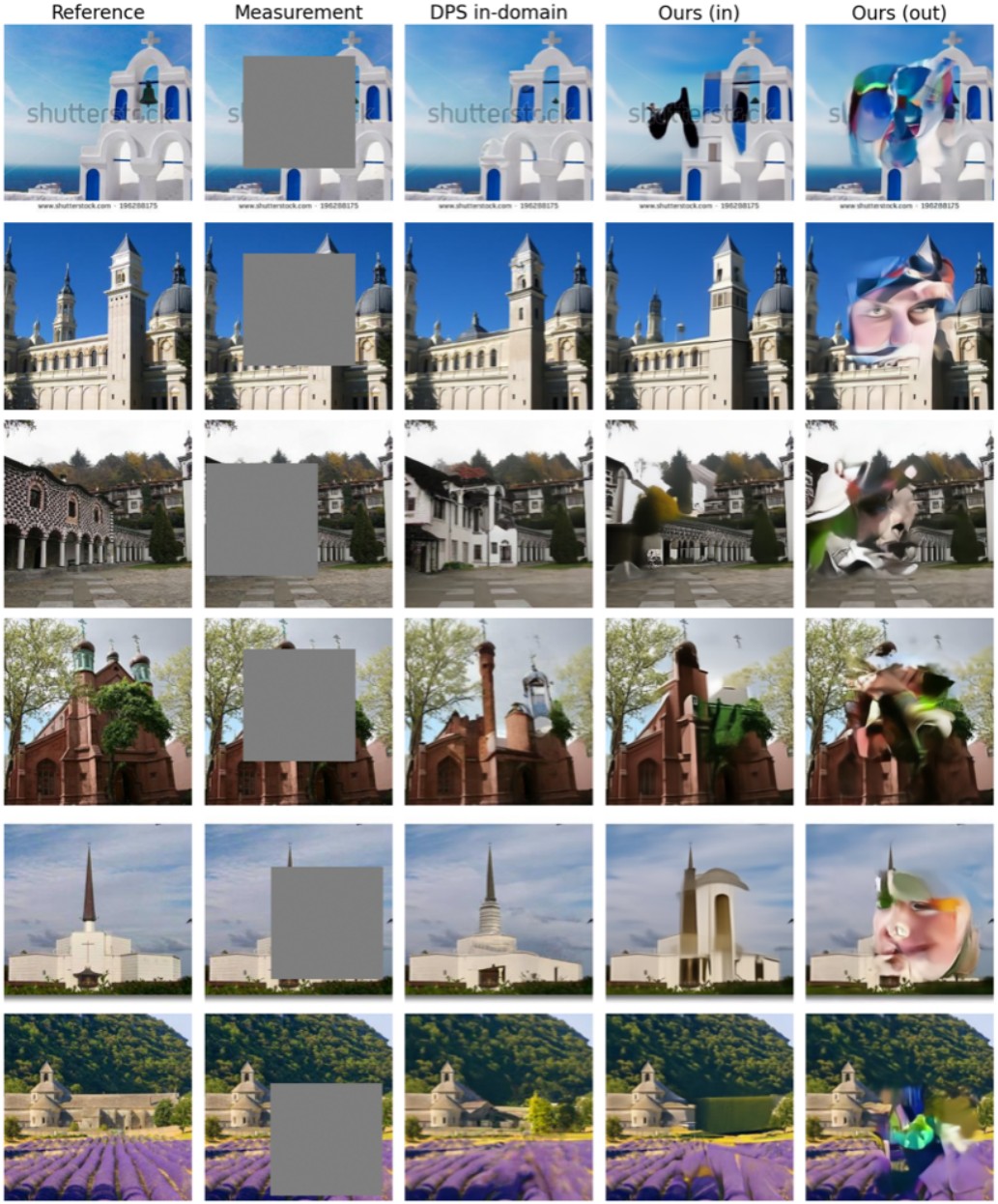

*Figure 14.* Sample qualitative reconstruction results for box inpainting with a $0.6 \times 0.6$ mask. DPS and Ours (in) uses Church pretrained model, Ours (out) uses CelebA pretrained model.

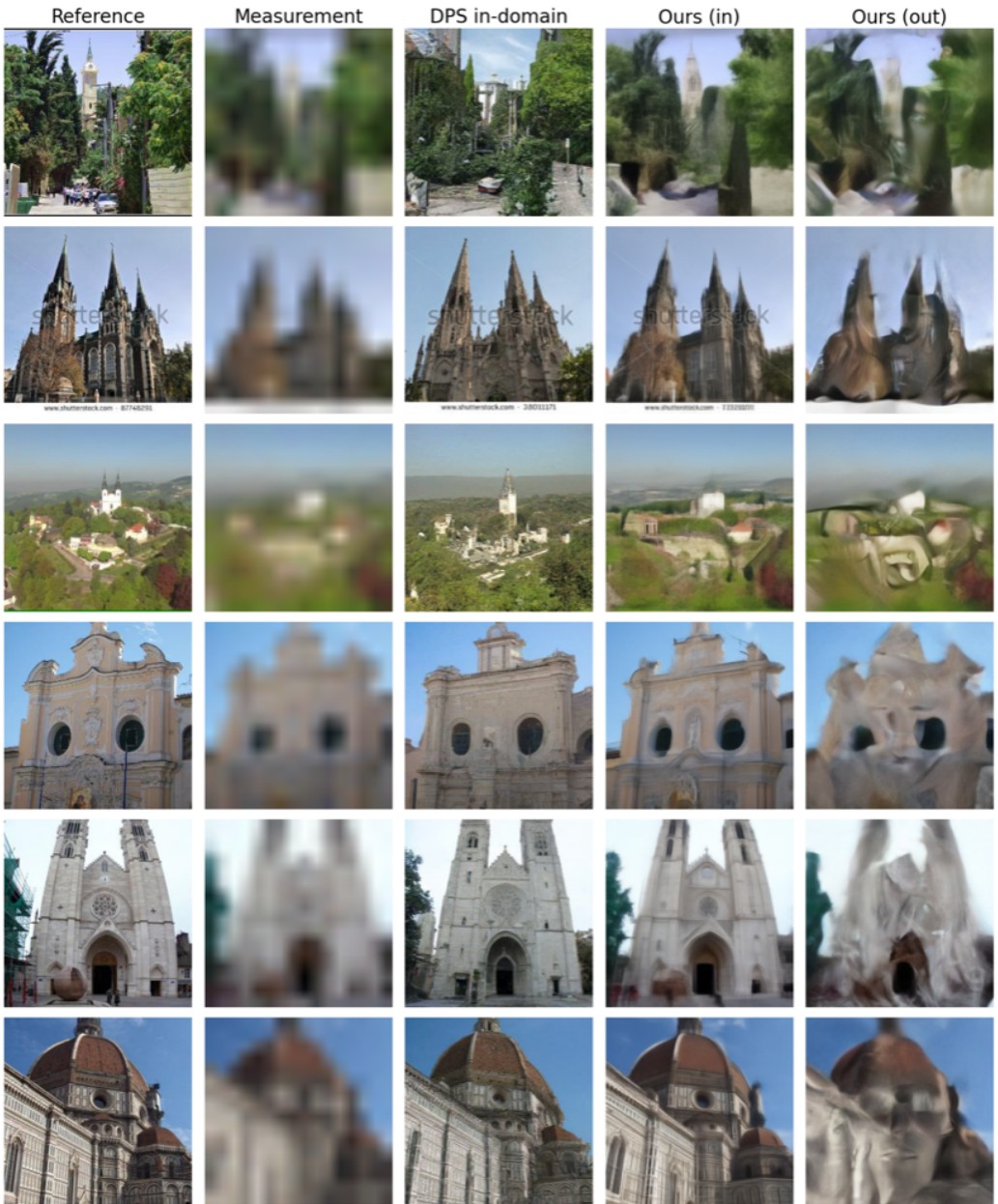

*Figure 15.* Sample qualitative reconstruction results for super-resolution with x16 ratio. DPS and Ours (in) uses Church pretrained model, Ours (out) uses CelebA pretrained model.

