# OpenReview forum: "Weak Diffusion Priors Can Still Achieve Strong Inverse-Problem Performance"
_ICML.cc/2026/Conference — ICML 2026 spotlight_

### Official Review · Reviewer_ath8 · 2026-02-20

**Soundness:** 3
**Presentation:** 3
**Significance:** 3
**Originality:** 3
**Overall Recommendation:** 4
**Confidence:** 4

**Summary:**

This manuscript's contribution concerns understanding when and why weak diffusion priors—either heavily truncated (few-step) or domain-mismatched models—can still yield competitive reconstruction quality in inverse problems. A notable domain studied by this article is diffusion-based inverse problem solving, specifically examining the robustness of reconstruction performance to prior quality through both empirical evaluation and Bayesian theoretical analysis.

The paper proposes that in data-informative regimes (high-dimensional, highly informative measurements), weak priors can match strong-prior baselines because the posterior is dominated by the likelihood rather than the prior. The authors support this with: (1) extensive cross-domain experiments showing 3-step DDIM priors (even out-of-domain) can match/exceed 1000-step DPS, (2) theoretical analysis via Gaussian mixture surrogates showing exponential posterior concentration under mild assumptions, and (3) algorithmic contributions (ADAMSPHERE optimizer, HOLDOUTTOPK early stopping) that stabilize initial-noise optimization.

**Compliance With Llm Reviewing Policy:**

Affirmed.

**Final Justification:**

The grade given - "weak accept" - is adequate for this paper and based on the provided response. I think that this paper desrves to be published, despite several weaknesses that resist in the fnal version.

**Key Questions For Authors:**

Beyond the obvious questions that arise from the above described weakness, here are few other questions that I found as disturbing:

1. Can you provide empirical evidence that the posterior actually concentrates on a single mode as predicted by Theorem 3.2 (e.g., by running multiple optimizations)?
2. How well does the Gaussian mixture surrogate approximate actual DDIM sample distributions? Can you quantify approximation error?
3. For the delta_0-identifiability assumption, can you extend the analysis in Table 1 to other tasks (deblurring, super-resolution) and mask ratios?
4. Why does HOLDOUTTOPK return the latest of top-K rather than the best? Is there empirical evidence this is better than selecting the single best holdout-error iterate?
5. The linear-M dependence in Theorem 3.2 could be prohibitive for large M. Is this bound tight, or is the actual dependence weaker?
6. Can you characterize domain mismatch quantitatively and predict performance degradation from a measure of domain distance?
7. For practitioners: given a pretrained model and target dataset, can you provide guidelines for predicting whether a weak prior will suffice?
8. How does performance vary with measurement noise level σ? Theory suggests higher noise needs stronger priors—does this hold empirically?

**Limitations:**

Yes

**Strengths And Weaknesses:**

Strengths

1. Important and Counterintuitive Observation: The finding that a 3-step bedroom-trained diffusion model can recover human faces (Figure 1) challenges conventional wisdom and has significant practical implications for data-scarce domains.

2. Comprehensive Empirical Validation: The experimental scope is impressive:
    o Multiple tasks (inpainting, deblurring, super-resolution)
    o Three datasets (CelebA, Church, Bedroom) plus ImageNet
    o Multiple architectures (pixel-space diffusion, latent diffusion with SD2.1 and DiT)
    o Cross-domain experiments systematically varying both prior strength and domain match
    o Tables 2, 5, and supplementary results show consistent trends

3. Solid Theoretical Framework: Theorem 3.2 provides a principled explanation via posterior concentration with clear assumptions. The Gaussian mixture surrogate is reasonable and well-motivated (Section 3.2), and the connection to classical posterior consistency (Theorem 3.1) grounds the high-dimensional analysis in established theory.

4. Thoughtful Failure Mode Analysis: The paper doesn't oversell—Section 4.3 clearly identifies when weak priors fail (box inpainting, large-scale super-resolution) and provides both empirical (Table 7, Figures 14-15) and theoretical justification (measurement informativeness too low).
5. Algorithmic Contributions: ADAMSPHERE and HOLDOUTTOPK are simple but effective refinements with clear ablations (Table 4) showing their value.
6. Clear Presentation: The paper is well-written with excellent visualizations. The progression from motivation (Figure 1) → theory (Section 3) → experiments (Section 4) is logical and easy to follow.

Major Weaknesses

Problem 1: If we are to compare priors, we should do so on the same grounds. Specifically, if we run DPS for 1000 steps with a matched prior, we should run the very same DPS with a mismatched prior in order to show that with sufficient measurements, we get almost the same results using two fairly equivalent restoration methods. This is not the strategy of this paper. Rather – the 1000 steps DPS is compared with the 3-DDIM prior that is injected into the optimization of the terms in Eq (1) or (2). This is a misleading comparison, because
- Eq. (1) or (2) lead to MAP estimation or maybe even to something that gets closer to MMSE estimation, whereas DPS strives to provide a posterior sampling outcome. Theoretically speaking, posterior sampling should give 3dB lower PSNR when compared to MMSE. Thus, getting better results with the 3-DDIM approach in PSNR (and LPIPS and SSIM) is not telling the whole story.
- DPS, providing posterior samples, is a stochastic solver. When activated many times, it can provide a cloud of solutions that approximate the posterior. All these solutions should be of high perceptual quality. In contrast, the solution of Eq. (2) (or (1)) is deterministic, and one that might tend to be blurrier.

Weakness of the prior in this paper refers to two very different scenarios – few steps diffusion injected as a prior into a MAP-like optimization task, or using regular diffusion restoration but with the ring denoiser. Mixing these two cases in the experiments is confusing, and should be avoided. Thus, Table 1 should be split into the two kinds of priors – a regular diffusion with 1000 steps (or less for ablation) with different denoisers deployed within, and a 3-DDIM prior and appropriate optimization for the varying denoisers. In all these comparisons, we should see FID as well, to assess whether the weaker prior hampered the perceptual quality of the results. One last thing: when using DPS for 1000 steps, it will be instructive to show the proximity of the outcomes to a fair sampling from the posterior – this could be done by generating a cloud of solutions for both options (suitable and mismatched priors) and evaluating the distance between these clouds somehow.

Problem 2: Theoretical analysis is lacking. Consider the pure Gaussian case and think of Tikhonov regularization. In a case of excessive measurements, the prior should be assessed in the null-space of the degradation operator, in order to claim that even an unmatched prior works well. More specifically, the claim of this work would be that such a null-space, that typically refers to fine-details and high-frequencies, is covered equally well by the true and mismatched prior. This is the desired strategy and yet it is very different from the overly simplistic modeling assumptions in the theoretical analysis within the paper.

Minor Weaknesses

• The authors should show the actual computational steps that give the 3-DDIM prior, to make it more concrete.
• Page 1, line 19: "bottom-left (resp. bottom-right)" - these directions don't match the figure layout
• Page 4, Eq. 3: Consider using π_prior(x) to distinguish from posterior π(x|y)
• Page 5: "tail@k" appears in Table 1 without definition—define in caption
• Table 2 caption: "target domainas" → "target domains"
• Figure 1 caption: Specify that "individually normalized" means per-image normalization to [0,1]
• Appendix A.4, line 840: "exponential map" retraction mentioned but not defined
• References: Several 2024-2025 arxiv preprints—verify publication status

---

> ### Author Rebuttal · Authors · 2026-03-30
>
> Thank you for the thoughtful and careful review! Below we respond to your two main questions and the additional comments. We’d love to hear your thoughts.
>
> ### Q1: Two different axes of “weak priors” and MAP vs sampling algorithms:
>
> Great point. In the paper, we study two axes of prior weakness: prior strength and domain match. DPS corresponds to the (strong, matched) setting, while initial-noise optimization was evaluated in the (weak, matched) and (weak, mismatched) settings. We agree that it is helpful to test DPS with a mismatched-domain prior (i.e. (strong, mismatched) setting). We also agree that MAP-like optimization methods and sampling-based methods should be distinguished more clearly.
>
> We have added new results for DPS with mismatched priors in **Table 8** of  https://anonymous.4open.science/r/weak-diffusion-priors-rebuttal-3E7F/README.md for inpainting and nonlinear deblurring. The message is consistent with our paper: domain mismatch slightly hurts performance, but does not destroy it. For example, in inpainting, DPS with a bedroom prior on CelebA still has PSNR above 30.6.
>
> In the revision, we will **present these two axes more cleanly, include and expand the DPS comparison with additional perceptual and distributional metrics, and include a brief discussion clarifying the different roles of optimization and sampling-based methods.**
>
> ### Q2: Null space analysis
>
> Thank you for sharing this perspective! The null space is indeed a natural place to assess the role of the prior.
>
> To better understand this point, we studied inpainting and $4\times$ SR and computed simple statistics, with results reported in **Table 10-11** of our repo. For SR, all priors place negligible energy in the null space relative to the observed passband, with ratios around $0.001–0.002$. For inpainting null space, the variance under a few-step sampler is much smaller than under the full sampler, which is expected since a 3-step DDIM produces blurrier images. We did an additional check by decomposing the reconstruction error into observed-space and null-space. For SR, the reconstruction error is indeed concentrated in the null-space component, whereas for inpainting the error is distributed evenly across both components.  Thus, we find the null-space diagnosis informative for e.g. super-resolution, but might not explain the full picture.
>
> Inspired by the suggestion, we further examined the spatial correlations of samples generated by priors of different strengths and from different domains. Results are summarized in **Table 12**.  We found they have nearly identical decay rates, with curves that are almost indistinguishable (corr $\ge$ 0.997). This suggests that they enforce very similar local structure, albeit differ in sample fidelity and semantic.
>
> Therefore, a **possible explanation** is that weak and strong priors of different domains share similar spatial correlation structure. The measurements provide local information, while the prior propagates it through spatial correlations to nearby locations and fills in the missing parts. **We will add further discussion in the revision.** We also view these experiments as a natural starting point for future work aimed at understanding the mechanism more carefully.
>
> ### Additional questions:
>
> **Q1: Optimization concentration.** Yes, given an image, we tried running the same optimization independently, the results are very close (Root mean square < 0.05).
>
> **Q2,5: Theory.** Gaussian mixtures are a standard surrogate model in diffusion theory. We use them since they give clean and interpretable results that make the Bayesian consistency mechanism transparent. We agree it is not exact, but we expect the Bayesian consistency message to remain valid under other reasonable models. The linear dependence on $M$ comes from the current proof, which may not be tight.
>
> **Q3: extending identifiability.** Empirically extending the analysis is simple. Theoretically validating using concentration inequalities like Prop A. 5 might be non-trivial.
>
> **Q4: Latest of top K?** We find when $K=1$, it slightly hurts PSNR and can hurt LPIPS more noticeably (~$10$%). We therefore choose $K > 1$ for greater robustness. **Table 4** in our repo has a small ablation study.
>
> **Q6,7: Quantify prior effect and practical guidance:** We provide additional statistics such as effective sample size as a metric for practitioners. We also propose an estimator to quantify the effect of different priors. Due to space limit, please refer to the response to **reviewer gJku** for details.
>
> **Q8**: Different noise level: We added an experiment in **Table 1-2** with noise ranging from 0.01 to 1. As noise increases, the gap between in-domain and out-of-domain performance indeed grows. When noise is very high, the performance of in-domain INO is worse than DPS, due to the weaker prior.
>
> **Minor issues:** will correct them all!
>
> Thank you again. We would be happy to answer any further questions and continue the discussion.

---

> > ### Author Rebuttal · Reviewer_ath8 · 2026-03-31
> >
> > The additional results address few of the concerns raised. That said, I still believe that the given grade is adequate.

---

> > > ### Author Response · Authors · 2026-04-07
> > >
> > > Thank you for your careful reading and constructive feedback.

---

### Official Review · Reviewer_gJku · 2026-03-08

**Soundness:** 3
**Presentation:** 3
**Significance:** 3
**Originality:** 3
**Overall Recommendation:** 4
**Confidence:** 2

**Summary:**

This paper studies an interesting and practically important question: when can weak diffusion priors still solve inverse problems well? The paper defines weak priors as either low-quality few-step samplers or priors trained on mismatched datasets, and argues that such priors can still achieve strong reconstruction performance when the measurements are sufficiently informative. Empirically, the paper evaluates random inpainting, Gaussian/nonlinear deblurring, super-resolution, and latent-diffusion-based ImageNet reconstruction. Theoretically, it provides a Bayesian-consistency-inspired analysis using a Gaussian-mixture surrogate prior, showing posterior concentration under a  $δ_0$-identifiability condition and large measurement dimension. The paper also proposes two algorithmic refinements, ADAMSPHERE and HOLDOUTTOPK, for initial-noise optimization.

**Compliance With Llm Reviewing Policy:**

Affirmed.

**Final Justification:**

I continue to maintain a positive overall assessment.

**Key Questions For Authors:**

1. **Does the proposed perspective lead to any concrete practical recommendation for method selection?**

    For example, when faced with a new inverse problem, when should one prefer a weak prior over a stronger full diffusion prior, and what trade-offs in reconstruction quality, robustness, or compute should one expect? If the authors can distill their findings into actionable guidance, I would view the paper as having greater practical impact rather than being primarily explanatory.

**Limitations:**

yes

**Strengths And Weaknesses:**

**Strengths**

1. The paper presents a highly novel and counterintuitive finding that weak or mismatched diffusion priors can still effectively solve inverse problems. It robustly supports this empirical observation with a solid theoretical framework based on Bayesian consistency, explaining how highly informative measurements can dominate the prior.
2. The proposed ADAMSPHERE optimizer and HOLDOUTTOPK early-stopping strategy offer valuable and lightweight solutions to stabilize initial-noise optimization and mitigate measurement overfitting.
3. The work is intellectually honest and complete, explicitly investigating and characterizing the boundaries and failure modes of weak priors.
4. The manuscript is well-written and easy to understand

**Weaknesses**

1. While the paper provides a systematic and elegant theoretical explanation for *why* weak priors can be effective, it falls short of translating these insights into tangible advantages for practical applications. The findings serve more as a retrospective diagnosis of an interesting phenomenon rather than a prescriptive tool that pushes the state-of-the-art forward.

2. The core premise of the theoretical framework relies on measurements being highly informative and the signal being $\delta_0$-identifiable. However, the paper lacks a practical, forward-looking metric to quantify this "information content" *before* running the algorithm. There is no clear way to determine *a priori* whether a specific degradation level (e.g., a exact mask ratio) crosses the threshold from "informative" to "uninformative". Consequently, determining whether a weak prior will succeed or fail (e.g., discovering the failure point at a $0.5 \times 0.5$ mask ) still requires trial-and-error empirical testing. This significantly diminishes the predictive value of the theory for guiding practical algorithm design.

---

> ### Author Rebuttal · Authors · 2026-03-30
>
> Thank you for the insightful question. We agree that an important next step is to turn the main message of the paper into practical guidance for method selection.
>
> We view this as two related questions: **(1)** given an inverse problem, when should we expect weak prior to work (roughly equally well with strong priors); and **(2)** given a prior, when should one use initial noise optimization (INO) rather than another solver e.g. DPS?
>
> ### Q1. When will weak prior work?
>
> We provide two sets of statistics. The first is essentially free, requires no extra compute, and can serve as an initial guide. The second is data-driven and requires some computation, but can be useful for larger-scale runs where this extra cost is acceptable.
>
> ### (a) Simple statistics
>
> We suggest reporting the following simple statistics.
>
> ### 1. Basic statistics
>
> The most direct quantities to report are:
>
> - the observed dimension $m$,
> - the input-to-output dimension ratio $m/d$,
> - the noise level $\sigma$.
>
> These are the simplest and most important summary statistics.
>
> ### 2. Noise-aware effective sample size
>
> For a linear measurement operator $A$ with singular values $s_i$ and noise level $\sigma$, define the noise-aware effective sample size and effective ratio by
>
> $$ r_{\mathrm{eff}}(A,\sigma)=\sum_i \frac{s_i^2}{s_i^2+\sigma^2}, \qquad \rho_{\mathrm{eff}}=\frac{r_{\mathrm{eff}}}{d}.
> $$
>
> Each term $\frac{s_i^2}{s_i^2+\sigma^2}$ acts like a soft indicator:
>
> - if $s_i \gg \sigma$, then the term is close to $1$, so that direction is measured reliably;
> - if $s_i \ll \sigma$, then the term is close to $0$, so that direction is dominated by noise.
>
> Our empirical heuristic is: for $256\times 256$ images, an effective sample size of at least $10{,}000$ (an effective ratio of at least $5$%) is a relatively safe regime for a 3-step/mismatched prior to perform well. We calculated these statistics for the experiments in our paper below. The reported PSNR, SSIM, and LPIPS also broadly follow the same ordering as these statistics, e.g., $70$% random inpainting is relatively the easiest task.
>
> | Operator | $r_{\mathrm{eff}}$ | $\rho_{\mathrm{eff}}$ |
> | --- | --- | --- |
> | Random inpainting | 58,976 | 0.300 |
> | Gaussian blur | 16,011 | 0.081 |
> | 4$\times$ super-resolution | 10,289 | 0.052 |
>
> ### (b) Data-driven Statistics
>
> Suppose there is a lot of images to be recovered, and we want to choose between two algorithms, which may use different priors. One simple strategy is to compare them directly at the reconstruction level. Given an observed $y$, we can fix the same noise $z$ and run  algos A, B with the same initial noise, and compare their average squared-distance $\sum_{i=1}^d ||\text{Algo A}(z,\text{prior A}\mid y)_i - \text{Algo B}(z, \text{prior B}\mid y)_i||^2/ d$ . We can then average this quantity over a small set of images to obtain a simple measure of the discrepancy between the two algorithms. If it is small e.g., $< 1$% of the pixel range, then we may treat the two algorithms as practically very similar. Mathematically, the above distance estimates an upper bound of the squared Wasserstein-2 distance between the distributions of $\text{Algo A}(z,\text{prior A}\mid y)$ and $\text{Algo B}(z,\text{prior B}\mid y)$. For example, we consider random inpainting on 10 randomly selected CelebA-HQ images, and compare the {3-step Bedroom prior, INO} vs {1000-step prior, DPS}. The average distance is $\Delta = 0.006$, which is very small.
>
> This method is more data-driven, but it requires additional computation. It may therefore be better suited to larger-scale settings, e.g., inpainting for a set of 10,000 images.
>
> ### Q2. When INO?
>
> INO and DPS differ both in goal and in how compute scales: INO optimizes one latent seed to fit the measurement, so its cost scales with optimization steps, while DPS runs a diffusion chain to approximate posterior inference, so its cost scales with diffusion steps
>
> In data-informative regimes, our experience is that INO can use extra computation effectively: it is often slower, but can achieve better reconstruction quality. For example, see e.g. **Table 2** of our paper and **Table 7** of https://anonymous.4open.science/r/weak-diffusion-priors-rebuttal-3E7F/README.md. Another advantage is that INO is simpler. It can also be adapted easily to other few-step generative models, e.g. GANs and flow-matching. Its drawbacks are cost and the fact that it is an optimization method rather than a posterior sampler. Therefore, it does not provide uncertainty quantification. In contrast, DPS-type algorithms are designed to be closer to posterior inference, at least in spirit.
>
> Our practical suggestion is that INO is most attractive when the measurements are sufficiently informative and one is willing to spend more computation for better reconstruction quality; stronger posterior-sampling methods become more attractive when uncertainty quantification is important or the problem is more weakly constrained.

---

> > ### Author Rebuttal · Reviewer_gJku · 2026-04-03
> >
> > The author's response has to some extent solved my problem, but I believe the current score is sufficient, so I will maintain the current score.

---

> > > ### Author Response · Authors · 2026-04-07
> > >
> > > Thank you for your positive feedback. We will include practical recommendations in the revised version.

---

### Official Review · Reviewer_49S6 · 2026-03-12

**Soundness:** 3
**Presentation:** 3
**Significance:** 3
**Originality:** 3
**Overall Recommendation:** 5
**Confidence:** 4

**Summary:**

This paper addresses a crucial topic in inverse problem solving: scenarios where there is no corresponding pretrained diffusion model available for a specific dataset. The authors conduct a comprehensive analysis of the mismatch between models and datasets across various inverse problem tasks, clearly identifying both the successes and limitations of utilizing weak or mismatched priors.

**Compliance With Llm Reviewing Policy:**

Affirmed.

**Final Justification:**

The authors have effectively addressed my initial concerns. The proposed method shows distinct advantages for solving scientific inverse problems under limited data constraints. A remaining weakness of the paper is that it understates the broader significance of this research direction. Nevertheless, I am recommending a '5: Accept' **only because** the authors have successfully incorporated results from scientific domains, and I consider this to be an important and promising direction for the field. However, I remain hesitant about this score because the authors did not include the scientific domain experiments in the actual paper. I defer to the Area Chair to judge whether the current level of empirical validation within the manuscript is sufficient for acceptance.

**Key Questions For Authors:**

Please address the concerns in the Strengths And Weaknesses

**Limitations:**

Yes

**Strengths And Weaknesses:**

This paper provides a series of insightful analyses of pretrained diffusion models in inverse problem solving. Personally, I appreciate this kind of work that breaks down the granular details of a model's successes and failure modes. However, I believe there are still several issues that the authors need to address.

1. In the Deep Image Prior (DIP) literature, there is a highly relevant work on Deep Random Projectors [1], which uses a randomly initialized neural network with a learnable seed to solve inverse problems. The method proposed in this paper shares significant similarities with it, if we consider a random network is a weak prior. It is important to include Deep Random Projectors as a baseline model and conduct a thorough evaluation and analysis.

2. The experiments in this paper focus solely on natural images. Given the availability of large-scale foundation models like FLUX and Stable Diffusion, restricting the evaluation to natural images diminishes the significance of this work, as weak/mismatched prior is generally not a major issue in this domain.

3. Scientific inverse problems offer a much larger stage to demonstrate the value of the proposed method. For instance, in InverseBench [2], the authors discuss prior mismatch and use a linear task to demonstrate the phenomenon. It would strengthen the paper significantly if the authors could extend the scope of their experiments to include scientific settings, especially when the task is nonlinear.

[1] Li, Taihui, et al. "Deep random projector: Accelerated deep image prior." Proceedings of the ieee/cvf conference on computer vision and pattern recognition. 2023.

[2] Zheng, Hongkai, et al. "Inversebench: Benchmarking plug-and-play diffusion priors for inverse problems in physical sciences." arXiv preprint arXiv:2503.11043 (2025).

---

> ### Author Rebuttal · Authors · 2026-03-30
>
> Thank you for your thoughtful review and for pointing us to the broader literature! The main questions are:
>
> 1. Deep random projector comparison;
> 2. Scientific inverse problems;
>
> We address the concerns and questions below with additional experiments and further discussion.
>
> 1. Thank you for pointing out Deep Random Projector (DRP)! We added a direct DRP comparison on random inpainting, and found that DRP is consistently 2–4 dB worse than our weak prior, with substantially lower SSIM.
>
>     We agree that DRP is a meaningful and closely related baseline, and we will add it to the discussion. In our framework, DRP can be viewed as an even weaker prior than those considered in the current paper. Unlike our weak diffusion priors, it uses a randomly initialized network and therefore does not rely on any learned data distribution, instead benefiting only from the architectural inductive bias of the generator. Since it uses a randomly initialized network, it is naturally not aligned with any target domain, and instead relies on the inductive bias of the network architecture to aid the inverse problem.
>
>     We have evaluated DRP on random inpainting on church, celebahq, and bedroom datasets, using the same setting described in Appendix C.6 of our paper. Results are shown in **Table 6** of https://anonymous.4open.science/r/weak-diffusion-priors-rebuttal-3E7F/README.md.  Compared with Table 2 of our paper, DRP gives reasonable results, but it is consistently 2–4 dB lower than our weak prior, and its SSIM is also much lower. We also show visual examples of DRP reconstructions in **Figure 2** of our anonymous repository. While DRP usually recovers the overall structure correctly, its facial features are less well defined, fine details are weaker, and the reconstructions are clearly blurrier. This is consistent with our intuition: DRP can offer useful structural regularization, but it does not use any learned data distribution, whereas our weak prior still carries useful information from pretraining, even when the prior is mismatched or low quality.
>
>     Therefore, we appreciate this suggestion and agree that DRP is a natural example of an even weaker prior that fits nicely within our framework. **We find these new results informative, and we will include them as well as comparisons on additional tasks in the revised version.**
>
> 2. Thanks for the suggestion! We added experiments on linear inverse scattering, and found that weak-prior initial-noise optimization remains competitive beyond natural images.
>
>     We fully agree that weak priors may also be useful in scientific inverse problems. Following this suggestion, we implemented linear inverse scattering in the same setting as Table 3 of [1]. We used the pretrained checkpoint provided in the repository, but replaced the original prior with a 3-step DDIM weak prior. We considered settings with 60 and 180 receivers; the results are reported in **Table 7**, with visualizations shown in **Figure 1** of our repo. With 180 receivers, the initial-noise optimization method is competitive, outperforming DPS and DDRM, although behind DAPS and RED-Diff. With 60 receivers, it achieves higher PSNR and SSIM than the existing methods. We also note that our method is slower than those in Table 3 of [1]. Therefore, we view this experiment mainly as a proof of concept showing two points: **first**, the weak-prior phenomenon can also arise in scientific inverse problems; **second**, initial-noise optimization can use additional computation to improve reconstruction quality in some regimes.
>
>     We plan to test on mismatched prior for scientific inverse problems in the future. We expect the same phenomenon still holds, for example, [2] shows brain MRI can still be used to reconstruct out-of-distribution scans such as knee MRI.
>
>     Thanks again for the suggestion. We find testing on scientific inverse problems to be a very good direction. **We will include these new experiments** **and discussions** in the revised paper, and we hope to report other interesting results along this line in future work.
>
>
> [1] Inversebench: Benchmarking plug-and-play diffusion priors for inverse problems in physical sciences, Zhang et. al., ICLR25
>
> [2] Robust compressed sensing MRI with deep generative priors, Jalal et.al., NeurIPS21

---

> > ### Author Rebuttal · Reviewer_49S6 · 2026-04-01
> >
> > I thank the authors for their comprehensive rebuttal and the substantial effort put into the new experiments. These additions effectively address my initial concerns regarding the similarity between this work and DRP.
> >
> > However, the weakness in a few-step DDIM prior stems primarily from the approximation error of the SDE—fewer steps naturally yield poorer reconstructions, which is fundamentally different from the prior mismatch. This specific dynamic has already been well-explored in works such as DMPlug.
> >
> > In contrast, I strongly believe the core value of your paper lies in its potential to address the mismatched prior. This is a highly practical issue for many scientific inverse problems where there is simply not enough data to train a domain-specific generative prior. While I appreciate that you acknowledge this in the rebuttal, deferring it to future work leaves a critical gap in the current manuscript. As noted in InverseBench (where direct optimization was evaluated), employing a mismatched prior can lead to slower convergence and poorer reconstruction quality, a point specifically highlighted in their *Appendix C: Robustness to prior mismatch*.I strongly suggest that the authors explicitly conduct a prior mismatch experiment using one or two datasets from InverseBench. Addressing this will significantly enhance the argument and practical value of your methods.

---

> > > ### Author Response · Authors · 2026-04-02
> > >
> > > We agree that prior mismatch is a practically important direction. Following the reviewer’s suggestion, we first focus on the Linear Scattering problem as a representative testbed for prior mismatch. We utilize the Full Waveform Inversion (FWI) prior provided by InverseBench, and evaluate across three distinct settings (60, 180, and 360 receivers) to simulate different levels of measurement informativeness.This FWI prior is severely mismatched in our setting, since it is trained for a very different task: recovering subsurface physical properties from full waveform measurements.
> > > In all settings, we compare DPS, DAPS, and Initial Noise Optimization (INO). All methods are lightly tuned. Results are given below:
> > >
> > > | Receivers | INO | DPS | DAPS |
> > > |---:|---:|---:|---:|
> > > | 60 | 21.97 | 20.29 | 21.33 |
> > > | 180 | 27.66 | 26.24 | 25.53 |
> > > | 360 | 27.99 | 26.46 | 26.68 |
> > >
> > >
> > > Compared with the strong-prior results in Table 3 of InverseBench, we find that a matched prior still gives the best reconstruction quality, as expected. That said, the performance drop under prior mismatch is not catastrophic, and it varies across inference methods. Reconstruction quality remains reasonable at 180 and 360 receivers; for example, at 180 receivers, the results are comparable to FISTA-TV and ΠGDM with strong priors. With 60 receivers, INO achieves a higher PSNR than FISTA-TV and several other diffusion-based methods that use strong priors recorded in Table 3. Among all the results that we have tested under mismatched prior, INO consistently outperforms both DPS and DAPS across all three receiver settings, with gains of 1.68, 1.42, and 1.53 dB over DPS, and 0.64, 2.13, and 1.31 dB over DAPS.
> > >
> > > We also evaluate a second InverseBench task: the nonlinear Black Hole Imaging problem. We use both a matched prior and a mismatched prior (Linear Scattering model resized to $64 \times 64$).
> > >
> > > Under the matched prior, DPS achieves the best PSNR, followed by INO and DAPS. Under prior mismatch, INO becomes the strongest method: it improves over DPS and DAPS by 2.04 dB and 2.82 dB in PSNR, and by 2.41 dB and 3.48 dB in blur PSNR. Moreover, INO shows the smallest degradation from the matched-prior to the mismatched-prior setting among all three methods.
> > >
> > > | Method   | PSNR (Mismatch) | PSNR (Correct) | Blur PSNR (Mismatch) | Blur PSNR (Correct) |
> > > | -------- | --------------- | -------------- | -------------------- | ------------------- |
> > > | **DPS**  | 18.06 ± 3.00    | 26.15 ± 4.82   | 21.98 ± 4.18         | 33.15 ± 6.87        |
> > > | **DAPS** | 17.28 ± 2.14    | 22.56 ± 3.81   | 20.91 ± 2.80         | 26.89 ± 4.55        |
> > > | **INO**  | 20.10 ± 2.93    | 24.15 ± 3.15   | 24.39 ± 3.87         | 29.32 ± 4.49        |
> > >
> > > We hope these results help address the reviewer’s concern about prior mismatch. We will include the additional results in the revision.

---

### Official Review · Reviewer_gTW8 · 2026-03-13

**Soundness:** 2
**Presentation:** 2
**Significance:** 2
**Originality:** 2
**Overall Recommendation:** 4
**Confidence:** 4

**Summary:**

This paper studies the feasibility of using weak priors and few-step diffusion models for solving inverse problems. The authors empirically find that when measurements are highly informative, even very weak priors can achieve reconstruction quality comparable to strong-prior algorithms. Additionally, the authors provide theoretical explanations for this phenomenon. Methodologically, based on DMPlug, the authors propose the ADAMSPHERE optimizer and HOLDOUTTOPK early-stopping strategy to stabilize the initial-noise optimization process.

**Compliance With Llm Reviewing Policy:**

Affirmed.

**Final Justification:**

All of my concerns were addressed.

**Key Questions For Authors:**

- I would like to know the quantitative comparison between the proposed algorithm and other algorithms in terms of time and memory costs. Although Figure 5 provides a comparison at the level of sampling steps, the gradient calculation process involved in this method may be more time-consuming than that of DPS.

- Can the authors provide additional comparison results against more diffusion-based algorithms?

- All experiments in this paper use Gaussian noise with noise level 0.01.  If at higher noise levels (e.g., 0.05 in DPS), does the viewpoint that weak diffusion priors can achieve reconstruction quality comparable to strong-prior algorithms in this paper still hold?

- The original text does not provide an analysis of the sensitivity of the hyperparameters such as learning rate and k for HOLDOUTTOPK. Can the authors provide relevant ablation experiments?

**Limitations:**

yes

**Strengths And Weaknesses:**

**Strengths**: The paper is well written and easy to follow. The idea of this paper is novel and of interest.

**Weaknesses**:

- The proposed method is rather similar to DMPlug, only differing in two components: ADAMSPHERE and HOLDOUTTOPK. Furthermore, Table 4 indicates that ADAMSPHERE alone shows limited improvement over the baseline, making it difficult to demonstrate its effectiveness.

- The proposed method uses 500 iterations to optimize the initial noise. If each iteration requires 3 sampling steps, this results in requiring 1500 sampling steps. Many diffusion solvers (such as DiffPIR and DDRM) require much fewer steps to reliably recover an image.

- The comparison methods in this submission are quite limited, only comparing with DPS and DMPlug, without comparing with advanced diffusion solvers such as DiffPIR and DAPS (and many others).

---

> ### Author Rebuttal · Authors · 2026-03-30
>
> Thank you for the thoughtful review! We agree it is beneficial to test whether our conclusions remain valid with (1) stronger diffusion baselines, (2) at higher noise levels, and (3) under explicit cost comparisons. We therefore ran new experiments addressing exactly these points, and the results continue to support the paper’s main claim: in data-informative regimes, weak priors can still produce strong reconstructions, often competitively with much stronger solvers. All new experiment results are provided at the following link: https://anonymous.4open.science/r/weak-diffusion-priors-rebuttal-3E7F/README.md.
>
> 1. **Additional Baselines**: We added additional comparisons with DAPS and DiffPIR on inpainting, Gaussian deblurring, and 4× super-resolution across CelebA-HQ, Bedroom, and Church. Results are recorded in **Table 5** of our repo. Our early-stopping results consistently achieve competitive PSNR. For example, in inpainting on CelebA-HQ, initial noise optimization (INO) achieves 33.78 PSNR, outperforming DAPS (32.35) and DiffPIR (31.96). In addition, out-of-domain results remain comparable. For instance, using a Church prior to recover CelebA still achieves 32.62 PSNR.
> 2. **Different Noise Levels:** Second, we tested higher Gaussian noise levels from σ = 0.01 to 1. The results support a more precise version of our claim: weak priors remain competitive at low to moderate noise, while the gap grows as the measurements become less informative. Results are provided in **Table 1-2**. For example, the out-of-domain prior remains competitive with DPS at σ = 0.05 and 0.20 (e.g., 32.08 vs. 30.12 PSNR at σ = 0.05, and 28.38 vs. 26.93 at σ = 0.20), but stronger priors become more helpful at very high noise. At the same time, as shown in **Table 2**, as noise increases, the gap between in-domain and out-of-domain performance grows. At the very high noise level $\sigma = 1$, DPS performs better. All these results are consistent with the main message of our paper: weak priors are most effective when the measurements are sufficiently informative.
> 3. **Ablation on K and learning rate:** We added ablations for the learning rate and HOLDOUTTOPK parameter k. INO is stable over a reasonable range: lr = 0.01 performs best, lr = 0.001 tends to underfit, and lr = 0.1 is less stable. For HOLDOUTTOPK, performance improves from k = 1 to moderate values and then becomes fairly insensitive. For example, choosing $k=50$ or $100$ typically improves PSNR, LPIPS, and SSIM simultaneously, with LPIPS improving by about $5-10$%.
> 4. **Additional compute match compute**: We agree that initial noise optimization has a higher runtime and memory cost than DPS. We now include several additional matched-cost comparisons in **Table 9**, using both wall-clock time and number of function evaluations (NFE). In a single full run, DPS with 1000 NFE takes 40.4 seconds and about 2.9 GB of memory, while initial noise optimization with 1000 optimization steps (3000 NFE) takes 171.2 seconds and about 8.5 GB. However, after normalizing by NFE, initial noise optimization at 333 steps (999 NFE) takes about 56.8 seconds and achieves 31.86 dB PSNR, which is close to DPS at 32.27 dB. At a larger matched budget of about 3000 NFE, initial noise optimization reaches 34.13 dB PSNR, compared with 32.36 dB for DPS using the best of 3 runs. Even against DPS using the best-of-5 runs (5000 NFE, 200.0 seconds, more compute than initial noise optimization), initial noise optimization still gives higher reconstruction quality, improving PSNR by 1.73 dB. This is in line with the intended use of INO as an optimization-based approach with a controllable cost-quality tradeoff.
>
> **Main message and Relationship to existing algorithms:** We would also clarify that these experiments are mainly intended to support the paper’s main message: weak diffusion priors can still solve inverse problems well when the measurements are sufficiently informative. In this sense, we view ADAMSPHERE and HOLDOUTTOPK as simple but useful components that help make the study stable and reproducible, rather than the main contribution of the paper. Likewise, we compare with DPS and other recent samplers because they provide strong baselines under the strong-prior setting, rather than to suggest a fully like-for-like comparison between methods of different algorithmic form.
>
> Thank you again for the constructive feedback, which has helped make the paper more complete and clearer! **We hope that our new experiments more fully illustrate the main phenomenon studied in the paper**: weak priors remain effective in data-informative regimes, stay competitive with stronger baselines at low to moderate noise, and continue to show a favorable quality-cost tradeoff under matched comparisons. **We will revise the paper accordingly and would welcome any further comments or suggestions.**

---

> > ### Author Rebuttal · Reviewer_gTW8 · 2026-04-03
> >
> > All of my concerns were addressed.

---

> > > ### Author Response · Authors · 2026-04-07
> > >
> > > Thank you for your positive feedback. We are happy to hear that our rebuttal addressed your questions.

---

### Decision · Program_Chairs · 2026-04-30

**Decision:**

Accept (spotlight)

**Comment:**

This paper studies the feasibility of using weak priors for solving inverse problems. Most reviewers agree that this paper is interesting and novel. The paper originally received 2xWeakReject and 2xWeakAccept. The main concerns include differences from DMPlug, limited comparisons, lack of theoretical analysis, etc. The authors have provided rebuttals and all reviewers mention that most of their concerns have been well addressed. Afterward, all reviewers lean to accept the paper. The authors are suggested to carefully revise the paper and incorporate newly conducted experiments according to the comments and discussions.